# A machine learning-enabled open biodata resource inventory from the scientific literature

Heidi J. Imker[1,2]*, Kenneth E. Schackart, III[1,3], Ana-Maria Istrate[4], Charles E. Cook[1]

**1** Global Biodata Coalition, Strasbourg, France, **2** University Library, University of Illinois at Urbana-Champaign, Urbana, Illinois, United States of America, **3** Department of Biosystems Engineering, The University of Arizona, Tucson, Arizona, United States of America, **4** Chan Zuckerberg Initiative, Redwood City, California, United States of America

* imker@illinois.edu

## Abstract

Modern biological research depends on data resources. These resources archive difficult-to-reproduce data and provide added-value aggregation, curation, and analyses. Collectively, they constitute a global infrastructure of biodata resources. While the organic proliferation of biodata resources has enabled incredible research, sustained support for the individual resources that make up this distributed infrastructure is a challenge. The Global Biodata Coalition (GBC) was established by research funders in part to aid in developing sustainable funding strategies for biodata resources. An important component of this work is understanding the scope of the resource infrastructure; how many biodata resources there are, where they are, and how they are supported. Existing registries require self-registration and/or extensive curation, and we sought to develop a method for assembling a global inventory of biodata resources that could be periodically updated with minimal human intervention. The approach we developed identifies biodata resources using open data from the scientific literature. Specifically, we used a machine learning-enabled natural language processing approach to identify biodata resources from titles and abstracts of life sciences publications contained in Europe PMC. Pretrained BERT (Bidirectional Encoder Representations from Transformers) models were fine-tuned to classify publications as describing a biodata resource or not and to predict the resource name using named entity recognition. To improve the quality of the resulting inventory, low-confidence predictions and potential duplicates were manually reviewed. Further information about the resources were then obtained using article metadata, such as funder and geolocation information. These efforts yielded an inventory of 3112 unique biodata resources based on articles published from 2011–2021. The code was developed to facilitate reuse and includes automated pipelines. All products of this effort are released under permissive licensing, including the biodata resource inventory itself (CC0) and all associated code (BSD/MIT).

**Data Availability Statement:** All code and data are archived in Zenodo along with associated documentation at https://zenodo.org/doi/10.5281/zenodo.10105161. The final inventory and its associated data dictionary are available as a

separate Zenodo deposit at https://zenodo.org/doi/10.5281/zenodo.10105947. Additionally, all materials are also available on GitHub at https://github.com/globalbiodata/inventory_2022/, which may be updated from the time of this publication. To facilitate exploration, an Open Science Products Table (S7 Table) provides the location of specific items.

**Funding:** This project was initiated by the Global Biodata Coalition as part of its programme of work, and which supported the work of CEC, KES, and HJI in planning and implementing the project. The Chan Zuckerberg Initiative supported the work of AMI in the development of machine learning methods.

**Competing interests:** The authors have declared that no competing interests exist.

# 1. Introduction

Scientists have long undertaken major infrastructure projects that examine large-scale scientific questions. Such initiatives often require sustained long-term support, frequently from a defined set of funders, for decades. The physical sciences have been especially adept at establishing infrastructures to produce data that are critically important for furthering scientific understanding. Typically, these projects, for example the large hadron collider at CERN and the James Webb Space telescope, are tangible structures or instruments that have well-defined physical locations (whether on earth or in space). Because they are tangible objects, the funders and the taxpayers who ultimately support the infrastructure can readily understand both how funds are spent and the necessity of long-term support in order to ensure that returns on the high initial investments are maximized.

In contrast, the life sciences' data infrastructure is highly distributed and largely virtual. There are thousands of distinct systems that provide access to structured biological data, collectively referred to as biodata resources. These resources archive difficult-to-reproduce data and also provide added-value aggregation, curation, and analysis to those archived data. Biodata resources are found throughout the world, vary in scale, and are supported by hundreds of funding bodies, institutions, and charitable foundations. This distributed infrastructure has grown dramatically over the past three or four decades as technological advances, such as in nucleotide sequencing, enabled exponential increases in the amount and types of data generated. However, and again unlike physical sciences, growth has been organic and driven locally. In the life sciences, individual researchers and institutions sought or provided funding to create each data resource, with new resources joining the infrastructure individually when they begin exchanging data with other resources.

The impact of biodata resources on life science research has been immense, and a number of efforts have been launched in recent years to improve their coordination and long-term sustainability. In Europe, ELIXIR was established in 2013 as an intergovernmental organization to coordinate life science data infrastructure. As part of ELIXIR's mission, they identified a set of European Core Biodata Resources that are of "fundamental importance" to research and show "wide applicability and usage" based on a set of quantitative and qualitative indicators. Initially, 19 resources were identified (now 22 after additional selection rounds), and literature mentions and citations show incredible reach. In an analysis of Europe PMC's full text articles, 17% were found to refer to a core resource [1].

Despite their critical importance for the life sciences research endeavor, biodata resources are usually funded precariously through short-term grants (generally 3–5 years) [1–3]. Research funders provide support for many of the biodata resources that comprise the biodata infrastructure, and they recognize both the need for long-term support of data resources and the challenges associated with creating long-term funding streams for such support [4]. While not all resources should live on in perpetuity, there is collective interest in establishing alternate funding mechanisms to stabilize the resources that make up the infrastructure [5]. In recognition of this challenge, research funders supported creation of the Global Biodata Coalition (GBC; globalbiodata.org) to aid them in coordinating funding for biodata resources and to develop mechanisms to more efficiently fund the biodata infrastructure. A basic requirement for coordinating support for this infrastructure is to understand its scope: how many biodata resources are there and where are they located? However, because biodata resources have been developed and managed independently of each other, this global overview is missing.

There have been many efforts to catalog biodata resources over the years. An early example is DBCat, launched in 1999 by the EMBnet branch Infobiogen, which used a combination of general web searches, journal review, and contributions from resource providers to assemble a

list of 511 resources [6]. Another effort in the biodiversity community found over 600 biodiversity information projects between 2005–2006 based on 100 hours of consulting effort [7]. There are also partial lists created by research funders [8], academic libraries [9], scholarly publishers [10], and Wikipedia [11]. The journal *Nucleic Acids Research* also maintains a catalogue of primarily molecular biology-related databases, the vast majority of which are described in one of the annual database issues of that journal [12].

Blair et al. [7] noted that catalogs quickly become outdated if static and are subject to funding and staffing challenges themselves. Indeed, the largest collection of molecular biology databases is Database Commons, hosted by National Genomics Data Center in Beijing China, which has developed its 5000+ record collection with contributions from over 50 curators. An alternative is encouraging resource owners to register their own resources. Currently, there are several options for such registration, including re3data.org [13], FAIRsharing [14], and the Sci-Crunch Registry [15]. All three actively encourage registration and include between ~ 1500 to ~ 3000 biodata resources, depending on interpretation of categories. Registration preliminarily happens in one of two ways; either a resource owner registers their resource themselves or another person, such as a curator at the registry, submits a record for the resource. The former path requires awareness of the registry and the latter path requires awareness of the resource, a loop that is a perennial challenge to close.

Given the interest from many different perspectives and the challenge of documenting the ever-growing life sciences infrastructure, we sought a method of assembling a global inventory of biodata resources and creating a process that would allow periodic updates. Here we describe the results of a reproducible, machine learning-enabled method to create this inventory by identifying biodata resources described in scientific articles between 2011–2021. The inventory developed, which contains 3112 resources, represents a use case-focused practical application of machine learning to address a question of interest to research funders and other stakeholders who support and use biodata resources across the globe. BERT (Bidirectional Encoder Representations from Transformers) models and the resulting metrics (i.e., F1, precision, and recall) were used in an exploratory manner to guide article classification and extraction of biodata resource names. To facilitate reuse, we released the inventory under CC0 licensing along with code available under BSD/MIT licensing; the code includes the machine learning steps in an automated pipeline plus scripts that extract value-add information about the identified resources from the metadata of associated articles. Along with a presentation of the methodology, we also provide a preliminary analysis of the inventory to demonstrate the potential for its reuse and augmentation.

## 2. Methods

### 2.1. Design overview

In order to create an open, reproducible inventory we needed a large source of open data that contains information about biodata resources and is structured enough to enable programmatic access. Text mining the scientific literature has been used to locate resources in previous studies, and Wren et al. combined this strategy with crowdsourcing to classify over 20,000 URLs extracted from MEDLINE abstracts, including 4757 that were designated as databases [16]. While Wren et al. did not identify resource names or locations, the high number of results suggested that scientific abstracts are a viable data source for identifying a large cache of resources. Furthermore, articles in centralized literature services are associated with high-quality metadata available via robust APIs. This means associated metadata, such as title, abstract, authors, author affiliations, funders, and citations, can be used to extract or infer information about the biodata resource itself, even when full text articles are paywalled. Additional

metadata associated with the resource URL extracted from the abstract (e.g., HTTP status and IP location) also allows collection of a set of useful characteristics for each inventoried resource. While there are limitations to this strategy since not all resource owners publish articles describing their biodata resources, many do. We hypothesized that we could not only create a large inventory that could be of use in and of itself, but freely releasing the inventory and the associated code would allow for reproducibility and extension by others. Given broad interest in the topic, we were particularly motivated by the idea that others may wish to subset or augment the inventory for other purposes.

Europe PMC is a large data resource of life sciences literature with an API allowing full access to the entire resource [17]. Using the strategy described above, we developed a targeted query to retrieve from Europe PMC a corpus of articles and then tested and used a machine-learning based approach to identify biodata resources named in this corpus (Fig 1). Openly available pretrained language models are currently state-of-the-art in natural language processing (NLP), achieving high performance on a variety of tasks such as Named Entity Recognition (NER), Question Answering, summarization, and machine translation. These models have also been adapted to the biomedical field by pretraining on domain-specific corpora (e.g., BioBERT, SciBERT, PubmedBERT). We defined two tasks: article classification, which aimed to classify a research article (based on the title and abstract) as being about a biodata resource or not, and NER, which extracted the exact mentions of biodata resources from text. We experimented with several of these pretrained language models relevant to the biomedical field by fine tuning them on these two tasks. We used a regular expression algorithm to extract URLs corresponding to the biodata resources and checked their HTTP response statuses and locations. With articles and their biodata resources defined, we then accessed additional metadata to further characterize individual resources, as mentioned above. This entire workflow was automated and made reproducible by implementing it in a Snakemake pipeline [18]. To maximize utility across a wide variety of potential users, the results have been made available as shown in the Open Science Products Table (S7 Table).

## 2.2. Open science implementation plan

At the start of the project, an Open Science Implementation Plan was created in order to clearly articulate our Open Science goals. The plan guided decision-making, allocation of time and resources, and the ultimate products of the project. The plan has four components: Reproducibility Standards, Code Standards, Data Standards, and External Review/Validation [19]. An account of our efforts to adhere to this plan, including a detailed description of steps taken to follow reproducibility standards outlined by Heil et al., are described in a companion article [20, 21].

## 2.3. Working definitions

Creating the inventory required definitions of "biodata" and "biodata resource" explicit enough to allow human curators to evaluate articles and create a training dataset for the machine learning models. For this inventory, which is intended to be of use to funders of basic research worldwide, we specifically excluded clinical and patient data resources that appeared to be aimed at clinical or diagnostic use rather than for use in research. We reviewed both formal ontologies and generic definitions related to the life sciences and the basic sciences (S1 Table) to assemble a working definition that reflected the objective of the inventory. The *life science biodata* definition used was "Biodata created through studies of living organisms and their associated life processes through research conducted for the specific purpose of acquiring fundamental knowledge; this knowledge forms the basis of testable theories that aim to

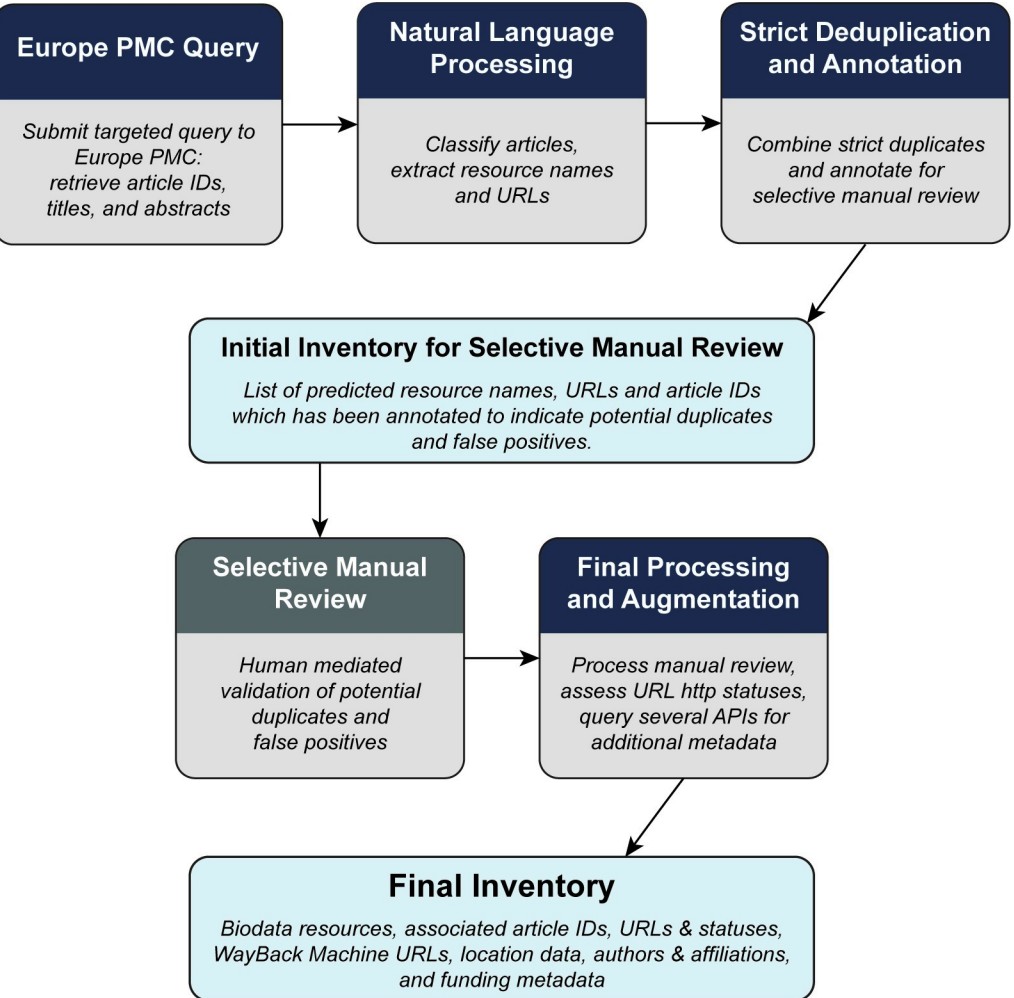

**Fig 1. Flowchart of overall study design to identify biodata resources from the scientific literature.** Study design for creating an inventory of biodata resources starting with a targeted query of articles found in Europe PMC, predicting biodata resources from article titles and abstracts using NLP, then reviewing and augmenting with additional metadata from Europe PMC to create the final inventory.

increase our ability to understand, interpret, and predict the phenomena that impact these organisms and processes."

To define "biodata resource" we likewise reviewed existing formal and general definitions (see S2 Table), including those from re3data.org, ELIXIR, the US National Institutes of Health, and the US Department of Energy as well as standards such as the Biomedical Resource Ontology and W3C's Data Catalog Vocabulary (DCAT). The *data resource* definition used was "An online source of structured data. The data cannot be a copy readily obtained from another resource (e.g., must be primary data or data annotated, curated, or otherwise augmented with value-added elements that are unique to the resource). The resource must have a distinct name and interface for browsing, searching, querying, viewing, and/or downloading the data within. Mechanisms such as an API may be the main access method, but there still must be a distinct online presence that provides information about the data available. Analysis tools may be provided, but information about and access to the underlying, unique data must be clearly available to users visiting the resource."

A particular challenge for this project was distinguishing biodata resources, which explicitly provide access to data, from tools that provide analysis or visualization of data, either through an inaccessible background database or via input by a user. In cases where the resource appeared to be both a source of biodata and a tool, it was included in the inventory.

## 2.4. Data sources and query development

To ensure that all project data are freely distributable and that the inventory may be updated programmatically in future, only open public data accessible via an API was used. The Europe PMC API was accessed using the Python requests library (v2.27.1) [22] to gather English-language articles that potentially describe a biodata data resource. The query for Europe PMC was iteratively developed to enrich the corpus with articles that potentially describe a biodata resource. Strings with various biodata resource-related terms and degrees of complexity were tested and the number and quality of results were manually inspected. On investigation, some initially promising terms were ruled out. For example, "repository" was often found in reference to code repositories. This resulted in a large number of false positives and created curation and classification challenges when associated with data analysis servers. We also looked at the impact of limiting the results to Open Access (OA) articles, which would enable more analysis later via the full text. However, restricting the query to OA returned less than half as many articles and was judged overly restrictive for the first step of the pipeline. In the converse, excluding terms to identify URLs returned millions of articles, only a very small fraction of which would describe a resource. Operators were also experimented with, and known true positives were used throughout as a check. After mid-project evaluation (see section 2.6 below), small adjustments to the query were made to exclude additional common false positives identified. The final query string was developed to locate abstracts that contained a URL and a small set of focused keywords that suggested the articles described a data resource while excluding retracted articles and those that include general URLs (e.g., for clinical trial registrations). The final query string used is shown in S1 Fig and provided in the project's data deposits and git repositories. To accommodate other queries, which may be better optimized or tailored to reflect a different use case (e.g., different year range, etc.), the pipeline developed calls for a user-provided query file, and tests in the code verify that the query passes successfully to Europe PMC.

Data retrieved from the re3data.org and FAIRsharing APIs are licensed for reuse and were used to benchmark the resulting inventory. Finally, Wayback Machine URLs and geo coordinates were retrieved via the Internet Archive Wayback Availability, ipinfo, and ip-api APIs as indicated in S3 Table.

## 2.5. Natural language processing (NLP) tasks

To generate the inventory from open data several natural language processing (NLP) methods were employed. ML models were trained to perform article classification and named entity recognition (NER) to identify articles describing biodata resources and extract their names. A regular expression was used to extract the URLs from the abstracts of predicted biodata resources. This workflow for the NLP tasks is shown in Fig 2, and details about the ML methods are covered in the section below.

**2.5.1 Training data.** To create the training dataset needed to develop a classifier via machine learning, a random sample of records returned from the query above was selected for manual review. The training set was created in two phases, with the first containing 638 records and second containing an additional 996 records for a total of 1634 records in the training dataset. Article titles and abstracts were independently reviewed by two curators in each phase

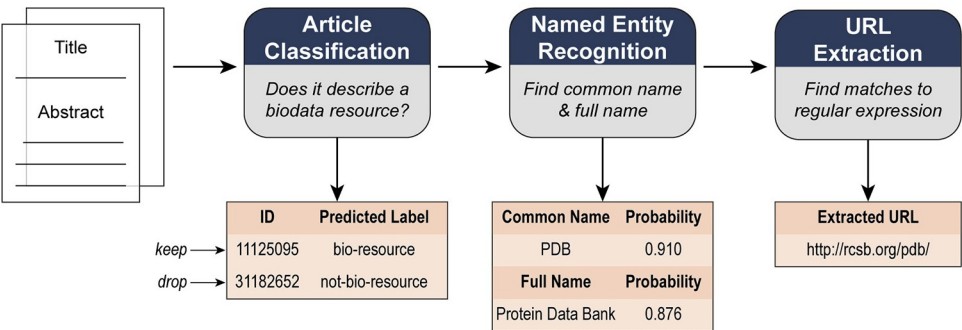

**Fig 2. Flowchart of the process used to extract mentions and URLs of biodata resources.** The article title and abstract are first fed into an article classification model, which determines if the article is about a biodata resource or not. Articles that are classified as being about a biodata resource are then fed into an additional Named Entity Recognition (NER) model which extracts the name(s) of the biodata resource and reports the average probability score of the tokens constituting the predicted name. A regular-expression based URL extraction algorithm separately extracts URLs from the abstract. In this example, the NER model extracted both a "full name" (Protein Data Bank) and a "common name" (PDB) for the resource, both of which are valid in this case.

and classified as either describing a biodata resource or not describing a biodata resource based on the developed definitions (see above). We kept the entries where both curators agreed on the article classification label (either positive or negative, n = 1587) and used this as a training dataset for the article classification task. For articles manually classified as describing a biodata resource, mentions of biodata resources in the title and abstract were identified, including "common names" (e.g., PDB) and "full names" (e.g., Protein Data Bank). This curated set of mentions was used for the NER task. Both training datasets were split into 70% training, 15% validation, 15% test (hold-out) for article classification (Table 1) and NER tasks (Table 2). Following common practice, the training sets were used to fine-tune the models on their respective tasks, the validation sets were used to compare the fine-tuned models for selection, and the test sets were used to evaluate how the models perform on unseen data [23].

**2.5.2. Models.**  Two machine learning models were fine-tuned to automate the process of 1) classifying research articles and 2) extracting mentions of biodata resources from those predicted to describe a biodata resource. Given a paper's title and abstract, the article classification model classifies the paper as being about a biodata resource or not. If an article receives a positive score, it is then passed through the NER model, which extracts the common name and full name of the biodata resource from the text, if they are present. A confidence score, computed as the average probability among the tokens constituting the mention, is also output (S2 Fig). BERT performs well on a variety of NLP tasks and several BERT derivatives have been pretrained on biomedical corpora, making them excellent candidates for this project. For both the article classification and NER task, BERT itself and 14 other BERT model variations available on Hugging Face Hub (https://huggingface.co/) were fine-tuned and evaluated to select the highest performing model (Table 3 and citations therein).

**2.5.3. Article classification task.**  The pre-trained models were fine-tuned on the article classification task. The model with the highest performance on the validation set was selected

**Table 1.  Training dataset splits for the article classification task.**

|  | Train | Validation | Test | Total |
|---|---|---|---|---|
| Positive labels | 337 (30.4%) | 61 (25.6%) | 80 (33.5%) | 478 |
| Negative labels | 773 (69.6%) | 177 (74.4%) | 159 (66.5%) | 1109 |
| Articles | 1110 | 238 | 239 | 1587 |

**Table 2. Training dataset splits for the Named Entity Recognition (NER) task.**

|  | Train | Validation | Test | Total |
|---|---|---|---|---|
| Articles | 306 | 66 | 66 | 438 |
| Biodata resource mentions | 1192 | 269 | 293 | 1754 |

and used to generate the inventory. In order to consider both title and abstract for classification the title and abstract were concatenated with a space character between fields to create a contiguous string. XML tags were removed using regular expressions, while adding white space after punctuation if not present after tag removal. The resulting input string was tokenized using a pre-trained tokenizer associated with the specific pre-trained model to be used for classification. Tokenized input was then passed through the pretrained model module to obtain context embeddings, which were subsequently fed into a linear classification layer that performs binary classification. This process classifies an article, based on title and abstract, as describing a biodata resource or not (Fig 3A).

Classification performance was evaluated on the validation set using precision (Eq 1), recall (Eq 2), and F1 score (Eq 3). For calculating performance metrics, an article that describes a biodata resource that is correctly classified is a true positive (TP), and if incorrectly classified is a false negative (FN). An article that does not describe a biodata resource that is correctly classified is a true negative (TN), and if incorrectly classified is a false positive (FP). Precision is the proportion of those articles predicted to describe a biodata resource that are indeed describing a biodata resource. Recall is the proportion of articles that describe a biodata resource that are correctly classified. F1 score is the harmonic mean of precision and recall.

$$precision = \frac{TP}{TP + FP} \tag{1}$$

$$recall = \frac{TP}{TP + FN} \tag{2}$$

$$F1 = \frac{2*precision*recall}{precision + recall} \tag{3}$$

**Table 3. Pre-trained models that were fine-tuned for the article classification and NER tasks.**

| Model | Hugging Face Model Name | Citation |
|---|---|---|
| BERT | "bert-base-uncased" | [24] |
| BioBERT | "dmis-lab/biobert-v1.1" | [25] |
| BioELECTRA | "kamalkraj/bioelectra-base-discriminator-pubmed" | [26] |
| BioELECTRA-PMC | "kamalkraj/bioelectra-base-discriminator-pubmed-pmc" | [26] |
| BioMed-RoBERTa | "allenai/biomed_roberta_base" | [27] |
| BioMed-RoBERTa-CP | "allenai/dsp_roberta_base_dapt_biomed_tapt_chemprot_4169" | [27] |
| BioMed-RoBERTa-RCT | "allenai/dsp_roberta_base_dapt_biomed_tapt_rct_500" | [27] |
| BlueBERT | "bionlp/bluebert_pubmed_uncased_L-12_H-768_A-12" | [28] |
| BlueBERT-MIMIC-III | "bionlp/bluebert_pubmed_mimic_uncased_L-12_H-768_A-12" | [28] |
| ELECTRAMed | "giacomomiolo/electramed_base_scivocab_1M" | [29] |
| PubMedBERT | "microsoft/BiomedNLP-PubMedBERT-base-uncased-abstract" | [30] |
| PubMedBERT-Full | "microsoft/BiomedNLP-PubMedBERT-base-uncased-abstract-fulltext" | [30] |
| SapBERT | "cambridgeltl/SapBERT-from-PubMedBERT-fulltext" | [31] |
| SapBERT-Mean | "cambridgeltl/SapBERT-from-PubMedBERT-fulltext-mean-token" | [31] |
| SciBERT | "allenai/scibert_scivocab_uncased" | [32] |

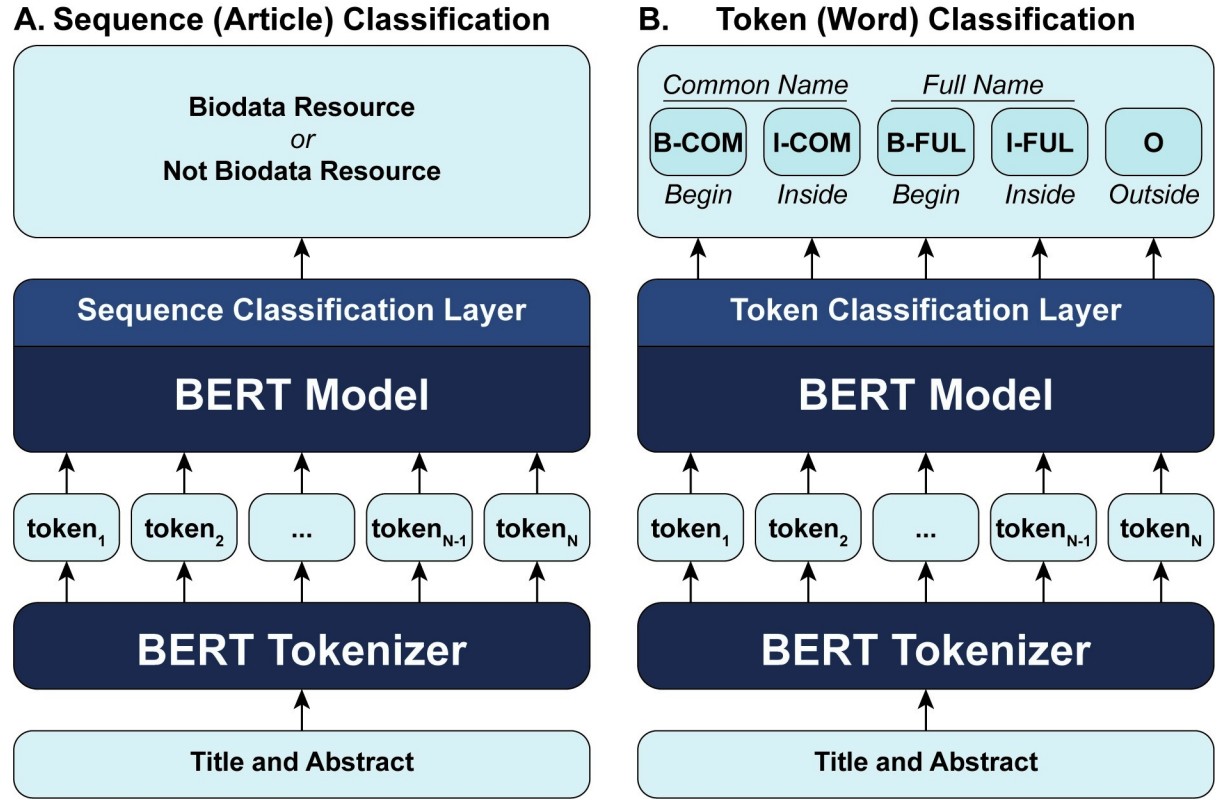

**Fig 3. Machine learning model architectures for article classification and named entity recognition.** Model architecture for (A) sequence classification used to classify articles based on title and abstract and (B) token classification used to perform NER to obtain the resource names. Each architecture shows the possible classification labels resulting from model prediction. Additional details for token classification are shown in S2 Fig.

All models were trained for a maximum of 10 epochs following general convention as a starting point. For each model the model checkpoint with the highest precision (regardless of epoch number) was saved. After evaluating the inventory in a mid-project evaluation (see section 2.6 below), precision was chosen over $F1$ and recall for the classification task to reduce the number of false positives. From this evaluation, we found that preferencing precision would improve overall quality of the inventory such that we would be able to finalize the inventory without manually validating every resource included. The precision scores of all the fine-tuned models on the validation set were compared to select the most performant model, which was then used for classification of unlabeled data.

**2.5.4. Named entity recognition (NER) task.** The same pre-trained models evaluated for the article classification task were fine-tuned for the NER task (Table 3). To accomplish this, the linear sequence classification layer was replaced by a token classification layer (Fig 3B and S2 Fig). Input (training and validation) data was tagged using the following BIO scheme [33]:

- B-COM: token is the start of a sequence corresponding to a common name

- I-COM: token is a non-start part of a sequence corresponding to a common name

- B-FUL: token is the start of a sequence corresponding to a full name

- I-FUL: token is a non-start part of a sequence corresponding to a full name

- O: outside

NER performance was evaluated using partial-match entity level metrics (precision, recall, *F*1 score) on the validation set. As with the classification task, models were trained for a maximum of 10 epochs. For each model architecture the trained model checkpoint with the highest *F*1 score was saved. The model with the highest *F*1 score on the validation set was then used for downstream tasks.

**2.5.5. Model implementation details.** The Huggingface framework was utilized to load the BERT model architectures [34]. The Huggingface's AutoModelForSequenceClassification module was used to fine-tune the BERT models on the article classification task, and the Auto-ModelForTokenClassification module to fine-tune the models on the NER task. The hyper-parameters used during fine-tuning of the models can be found in S4 Table. We used the Adam optimizer for training and the seqeval module for computing partial match entity-level metrics. Models were trained for a maximum of 10 epochs. All machine learning code was implemented in Python v3.8.12 and used the following third-party packages: Datasets v1.18.3, Natural Language Toolkit (NLTK) v3.6.1, NumPy v1.19.2, Pandas v1.2.4, pytest v6.2.4, scikit-learn v0.24.1, PyTorch v1.9.0, tqdm v4.63.0, and transformers v4.16.2 [34–42].

## 2.6. Mid-project evaluation and iteration

We conducted a manual evaluation midway through the project to assess how well the predictions were performing in practice and if any improvements could be made before we continued. At that point, SapBERT resulted in the highest best *F*1 score of the 15 models tested for the classification task (S5 Table). Using the results of that model, a curator assessed precision on a 10% random sample (n = 468) of the predictions to assess the correctness of the article classifications and NER extracted terms. Classification was determined to be either correct or incorrect. For the NER outputs, the evaluation determined if the extracted common and full names were correct, partially correct, or incorrect. This step helped us confirm that machine learning was indeed viable for this project and also helped us determine that a selective manual evaluation of low-scoring predictions would still enhance the overall quality of the inventory. To determine which predictions qualify as "low-scoring" for this first inventory and updates in the future, we used the results of the midway evaluation to determine a threshold probability of < 0.978 as "low-scoring," where 0.978 was the average probability for names determined by a curator to have been correctly predicted in the 10% random sample of the 468 articles that were manually reviewed (see additional details in Results section 3.3.1 Mid-Project Manual Evaluation). Therefore, any resource whose highest scoring predicted name ("best name," regardless if common or full) has a probability below this threshold is annotated to be manually reviewed by a curator before the inventory is finalized. In addition to the threshold, this step also helped us determine what iterative improvements could be made to refine the overall strategy, such as revising the input query (e.g., excluding "onlinelibrary.wiley.com" URLs) and adding conditionals to the models (e.g., predicted name length > 1 and number of URLs $\leq$ 2; see Section 2.7.6.).

During preparation of this manuscript, we realized that a portion of the ML test sets may have been present in the set of articles that were manually reviewed during the mid-project evaluation. While we did not pass labeled data through the pipeline, we did process all of the articles returned from Europe PMC. The potential impact of such data leakage was assessed and is reported in Results.

## 2.7. Post processing

**2.7.1. Best name determination.** As detailed above, for each article the NER model attempts to predict both full and common names and may output multiple predictions, each

of which is associated with a probability score, where the higher the score the greater the confidence in the predicted name. To determine the highest quality names, the probability scores for named entities of each type were compared to determine the "best common name" and "best full name". These probability scores of these two named entities were compared to choose the best overall name "best name" (that with the highest probability score).

**2.7.2. Automated deduplication.** Many resources publish repeatedly, for instance, to provide updates about the resource. Consequently, the raw inventory contained duplicate records from several articles describing the same resource. A first step toward deduplicating the inventory was performed by identifying records that had the same predicted best name and same extracted URL (ignoring differences due to trailing slashes or "http:" vs "https:"). These duplicate resources were merged with the PMIDs of each original article and retained along with the title-abstract text and publication date of the most recently published article.

**2.7.3. Annotation for selective manual evaluation.** Up to this point all steps were automated. However, based on the results of the mid-project evaluation described above, we realized that it would be advantageous to conduct a selective manual evaluation of some predicted resources to improve the overall quality of the inventory. In preparation for this step, a script within the pipeline added a new column with the variable "low_prob" for any resource whose best name probability < 0.978, the value determined in the mid-project evaluation as the average of correctly predicted names in the 10% random sample. This served as a flag to aid the curator conducting the manual evaluation. Additionally, while the automated deduplication was able to merge any articles with exact names and exact URLs matches, we were aware of suspected duplicates (e.g. variable names such "FANTOM" and "FANTOM5" sharing the same extracted URL while variable URLs such as http://appris-tools.org and http://appris.bioinfo.cnio.es share the same predicted name "APPRIS"). Deduplication on either name or URL (as opposed to both) would have led to erroneous mergers. For example, "Seed" and "SEED" are two different resources, while there are at least three distinct "PED" resources and two distinct resources for "SMART." Instead, to account for potential duplicates, columns were also generated for matching best names (flagged with "duplicate_names") or matching extracted URLs (flagged with "duplicate_urls") for evaluation by the curator. While more complex, automated procedures may be warranted in the future as the inventory grows over time, these cases were relatively few and we judged this strategy to be sufficient for the time being.

**2.7.4. Selective manual evaluation procedure.** With steps implemented above, a preliminary inventory with records flagged for manual review was generated as a CSV file. A curator then reviewed each flag to determine if low probability records should be removed from the inventory and if potential duplicate records should be merged within the inventory. This review was done in Microsoft Excel, with data validation applied to a set of predetermined outcomes (e.g. "remove", "merge", etc.). Importantly, no corrections to the predicted names or URLs were made; thus all values within the inventory are the output of the machine learning pipeline, which reduces the confusion that could result if the inventory were a mixture of ML-generated and human-generated names/URLs (especially in future updates). Review guidelines were developed to help standardize handling of edge cases (see S7 Table). The resulting manually reviewed file was added into the directory for subsequent processing. The pipeline first ensures all flagged records contain only valid review values and then removes or merges the appropriate records before moving on to metadata augmentation.

**2.7.5. Metadata augmentation.** Once we had the final prediction script and the results of the manual review processed, the Europe PMC API was once again queried using PMIDs to retrieve author affiliations, author names, grant IDs, grant agency name, and citation counts (i.e. via metadata elements 'affiliation', 'fullName', 'grantID', 'agency', 'citedByCount', respectively) for each article associated with the biodata resources.

**2.7.6. URL processing.** Biodata resources may be impermanent for reasons that include loss of funding, loss of key personnel, and technological change leading to deprecation, and we were interested in establishing if the URL provided in the abstract was still viable. Accordingly, the extracted URLs were checked for viability through standard HTTP status calls using the Python requests and urllib3 (v1.26.8) [43] libraries, and the returns (e.g. 200 OK, 404 Not Found, etc.) were recorded in the inventory on 11 November 2022. Extracted URLs were tested three times, with each attempt allowing 5 seconds for a response. The second attempt is submitted immediately after the first, while the third attempt is submitted after a one second delay.

Web archives offer a chance to locate snapshots of previously available websites [44]. To mitigate current and anticipate future availability issues, we used the Internet Archive's Wayback Machine API to check URLs for the presence of archived sites. While the Wayback crawler is often unable to access the data itself, these snapshots provide views of HTML pages such as the home page, search interface, etc. which provide important context in the absence of the live site. For the biodata resources in this inventory the most recent Wayback Machine URL was recorded for successful returns; for live URLs not represented in the Wayback Machine the URLs were submitted for archiving and the associated Wayback URL recorded. While the Wayback Machine is able to crawl and archive the majority of sites represented here, sites behind firewalls or those that prohibit crawlers cannot be archived.

**2.7.7. Resource geolocation.** We use two methods to identify the location of the biodata resource. The first is the location as determined from the IP address, which suggests a physical location for the infrastructure. For those URLs that return a status less than 400, the IP address is obtained from the host name. In an attempt to geolocate the IP addresses, ipinfo [45] and ip-api [46] are queried to request the IP address country and coordinates. Two APIs are used since neither is complete, and querying multiple APIs increases the chance of successful geolocation. When a location is successfully obtained from either API, the country and coordinates are recorded.

Because of the global nature of the life sciences research enterprise, physical location alone may not reflect collaboratively developed resources. Therefore, we also extracted country names following ISO 3166 from the author affiliations available from the Europe PMC metadata. When geolocating IP addresses both coordinates and country name were returned. ISO-3166-1 Alpha-3 codes and country names of all countries were searched against both the IP address geolocations and author affiliations [47] and recorded in the inventory.

## 2.8. Workflow management

The Snakemake workflow manager is used to automate the process described above in two pipelines. The first pipeline performs data splitting, model training (including model selection and evaluation), prediction, and downstream processing prior to selective manual review. After the selective manual review this pipeline resumes for final processing. This process, excluding model training and selection, is shown in Fig 1. A second pipeline was also developed to facilitate updating the inventory in future using new queries to Europe PMC and the previously fine-tuned models.

## 2.9. Analysis of the final inventory

With the final inventory established and additional metadata elements added, we carried out analyses on location, funders, and text-mining related metadata to provide a preliminary analysis of the resources identified and identify future opportunities to further explore and reuse the inventory. Using the geolocation information (see Section 2.7.7. above), location information was mapped without additional cleaning or filtering. All analyses were performed in R

using the packages argparse, dplyr, europepmc, forcats, ggplot2, glue, gt, httr, jsonlite, magrittr, purrr, RColorBrewer, readr, scales, stringr, tibble, tidyr, and xml2 [48–67]. The data resulting from these analyses, as well as the scripts to perform them, are available in GitHub and archived in Zenodo along with the code and data for developing the inventory.

**2.9.1. Analysis of funders.** To evaluate the funding agency names found in article metadata, agency names which appeared three or more times were extracted and the countries of origin for the funding body were manually identified through Google searches. Recorded country names were verified by a second curator and then standardized to a three-letter coding following ISO 3166–1 alpha-3. For funders international in nature, which are common for funding provided through the European Union, a unique 3-letter code of "INT" was used in the absence of an ISO standard.

**2.9.2. Comparison with existing registries.** To get a sense of how well this machine-learning enabled method of identifying biodata resources may complement other methods, we compared the biodata resources in this inventory with life science data resources collected elsewhere. There are several collections, such as the catalogue maintained by the journal *Nucleic Acids Research* and Database Commons; however, neither offer bulk download or programmatic access to their data. As such, we focused on two registries, re3data.org [13] and FAIRsharing [14]. Both carefully curate their registered resources and provide robust API access. These registries, which were created for different purposes and via different teams, do not themselves contain all known data resources but inclusion criteria are similar to the inventory (see below). Therefore, we anticipated some overlap but not a complete union with either.

Library and information science professionals in Germany established re3data.org in 2013, specifically as a registry of research data repositories, and the resource has maintained a domain agnostic collection policy. The criteria for inclusion in re3data requires that a resource 1) be run by a legal entity, 2) has access conditions and terms of use, and 3) focuses on research data. We accessed all resource records from re3data's API and then filtered to align with the inclusion criteria for the inventory. To do this, we subsetted to only resources categorized as "life sciences" which had URLs and removed the generalist categories (specifically "institutional" or "other") to focus on resources specifically dedicated to the life sciences. FAIRsharing emerged from its more focused predecessor, BioSharing, itself originating from the Minimum Information about a Biomedical or Biological Investigation Portal. Along with data resources, FAIRsharing also includes standards and policies and has expanded beyond the life sciences. FAIRsharing explicitly excludes individual datasets, consistent with the inclusion criteria of the inventory, and requires that resources are 1) organised collections of data, 2) findable via an active website that allows users to browse and/or search, and 3) accessible regardless of specific license type. We retrieved only resource records for the life sciences category from the API and did not filter the results further.

To prepare the resources from re3data, FAIRSharing, and the inventory for comparison, white space was trimmed from the resource name, and resource URLs were cleaned by removing the scheme and any trailing slashes and then converting to lowercase. As noted above, resource names and URLs may vary slightly but significantly, so we compared by common name, full name when available, and URL separately, merged the results, and then deduplicated where both the name and URL matched to obtain the final number of resources found in common.

## 3. Results

### 3.1. Overview

We initially retrieved 21716 articles from Europe PMC using all data sources, but later restricted this to the 20880 which had PMIDs available since we found that the metadata

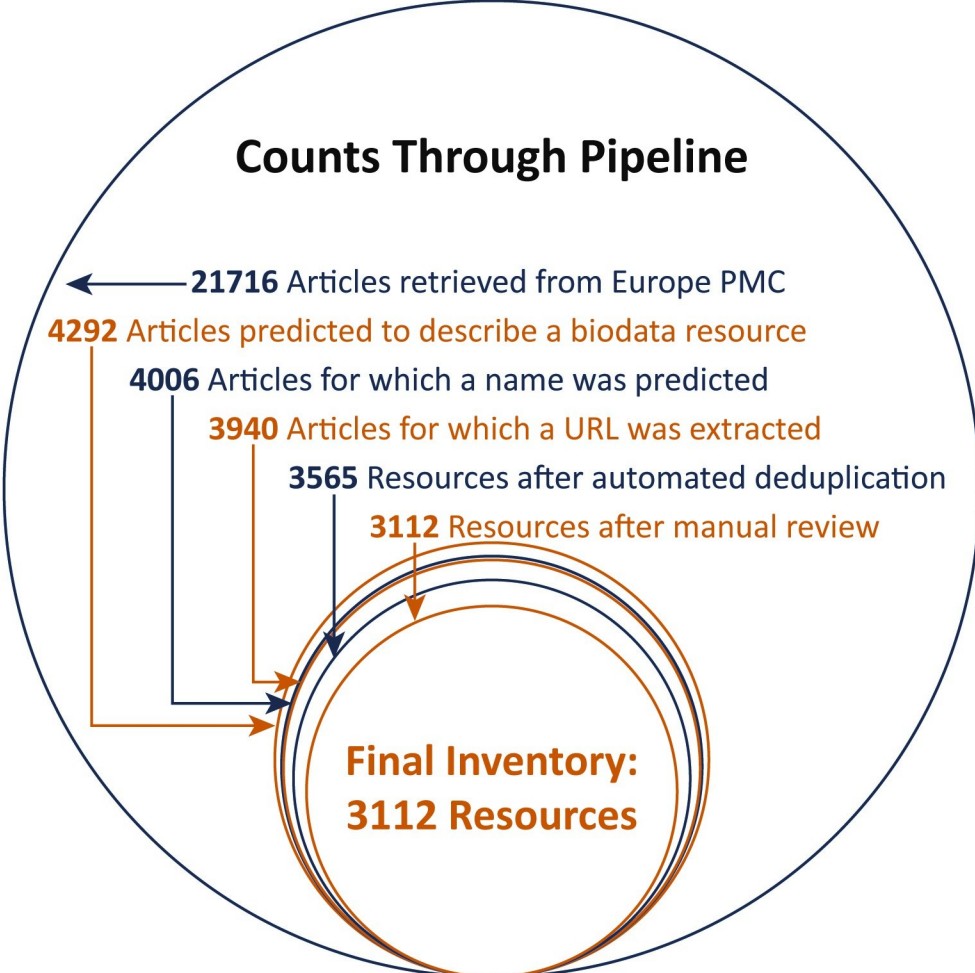

**Fig 4. Counts of articles and resources as they passed through the pipeline.** The area of each circle is proportional to the count at that step. Most attrition occurs during the initial classification step that predicts which articles describe a biodata resource, while losses are relatively low as additional steps build and finalize the inventory.

associated with articles lacking PMIDs was often insufficient for further analyses. During the article classification task the model predicted a negative label for 16588 articles while 4292 articles were predicted to describe a biodata resource. Of those with a positive label, the NER model extracted at least one name (common or full) for 4006 articles. Those without at least one complete URL were removed, resulting in 3940 articles. After deduplicating articles that had the same "best name" and same extracted URL, 3565 potential resources were identified in the preliminary inventory. After selective manual review to evaluate and remove erroneous low probability resources and merge additional duplicates, the final inventory contains 3112 biodata resources (Fig 4).

### 3.2. Training data and NLP tasks metrics

During curation of the 1634 records in the training dataset, initial inter-annotator agreement was high (89.4%) and increased to 97.1% once conflicting scores were reviewed, discussed, and reclassified. The training set took a total of 33 curation hours. While curation was relatively quick and straightforward, challenges fell into two main categories where it was difficult to

distinguish between biodata resources and other resources that 1) are available solely as a tool or 2) belonged to a different disciplinary area (such as clinical health records).

For each of the article classification and NER tasks we divided the training datasets into 70/15/15 splits used for training, validation, and testing, respectively. Models were evaluated using Precision, Recall and the *F*1 scores. All pretrained models were fine-tuned and evaluated using the same training and validation sets. Models with the highest performance when run using the validation data were chosen for final implementation. Precision was used for comparing the article classification models, and *F*1 score was used for comparing the NER models. Performance of all fine-tuned models on both validation and test sets are provided in S5 and S6 Tables. For the article classification task BioMed-RoBERTa-RCT had highest precision on the validation set and had a precision of 0.975, *F*1 score of 0.821, and recall of 0.719 on test data that had not been seen during training. For the NER task, BioMed-RoBERTa-RCT had the highest *F*1 score on the validation set and had an *F*1 score of 0.717, precision of 0.689, and recall of 0.748 on test data.

## 3.3. Manual evaluations

In addition to NLP metrics above, we also manually evaluated results at two points in the project. To assess the viability of application of machine learning to this project and to determine if any improvements could be made to the strategy overall, precision was determined for a 10% random sample of preliminary results mid-way through the project. The second evaluation was performed prior to finalizing the inventory. This evaluation reviewed resources with low probabilities and suspected duplicates. The results of both evaluations are detailed below.

**3.3.1. Mid-project manual evaluation.** In the mid-project evaluation of a 10% sample, 468 articles were manually reviewed. Of these, we found that 439/468 (0.938) articles were classified correctly. In Fig 5A, we show an example of a correctly classified article which was relatively straightforward while in Fig 5B, the text describes an entity that appears, on human reading, to be a tool that does not provide access to data [68, 69]. Another scenario that proved challenging for the models was when the title-abstract described a research project for which data was deposited into a resource (such as Flybase [70]) but the article did not describe the resource itself. However, overall these errors were limited, and given the challenge of classification even for human curators, we judged that the machine learning based methodology was indeed viable for classification.

In addition to evaluating the correctness of classifications, in the mid-project evaluation we looked at the results of the NER extraction for all 468 articles of the 10% sample to determine if extracted common and full names were correct, partially correct, or incorrect. A common name was predicted for 457 of the 468 (97.6%) in the sample, and in 425 of these predictions (93.0%), the common name with the highest probability was, in fact, the correct common name. Notably, even when not correct, only 10 (2.2%) were entirely incorrect (e.g., predicted as "the" instead of "WikiPathways Database") while 22 (4.8%) were partially correct (e.g., predicted as "Open" instead of "Open TG-GATEs"). In reality, for several that were entirely incorrect, the name was so poorly articulated in the abstract that it was difficult for a human to determine a valid name. Full names were predicted less frequently on the whole (157/468, 33.5%) and, reflective of greater complexity (see Fig 6 for an example), were less likely to be judged correct with 97 being correct (61.8%). However, only 7 (4.5%) were entirely incorrect and 53 were partially correct (33.8%). From these results, it was clear to us that, as was true with predicted classification, the method was viable for predicting resource names, with common names being the most likely identified.

While it was clear from the results of the mid-project evaluation that the machine learning methods would be a useful technique to employ, we also recognized that the quality of

## A) Correct Positive Classification

**PMID** 30942868 **Title-Abstract:** Digital expression explorer 2: a repository of uniformly processed RNA sequencing data. Background RNA sequencing (RNA-seq) is an indispensable tool in the study of gene regulation. While the technology has brought with it better transcript coverage and quantification, there remain considerable barriers to entry for the computational biologist to analyse large data sets. There is a real need for a repository of uniformly processed RNA-seq data that is easy to use. Findings To address these obstacles, we developed Digital Expression Explorer 2 (DEE2), a web-based repository of RNA-seq data in the form of gene-level and transcript-level expression counts. DEE2 contains >5.3 trillion assigned reads from 580,000 RNA-seq data sets including species Escherichia coli, yeast, Arabidopsis, worm, fruit fly, zebrafish, rat, mouse, and human. Base-space sequence data downloaded from the National Center for Biotechnology Information Sequence Read Archive underwent quality control prior to transcriptome and genome mapping using open-source tools. Uniform data processing methods ensure consistency across experiments, facilitating fast and reproducible meta-analyses. Conclusions The web interface allows users to quickly identify data sets of interest using accession number and keyword searches. The data can also be accessed programmatically using a specifically designed R package. We demonstrate that DEE2 data are compatible with statistical packages such as edgeR or DESeq. Bulk data are also available for download. DEE2 can be found at http://dee2.io.

## B) Incorrect Positive Classification

**PMID:** 24627222 **Title-Abstract:** miR-Synth: a computational resource for the design of multi-site multi-target synthetic miRNAs. RNAi is a powerful tool for the regulation of gene expression. It is widely and successfully employed in functional studies and is now emerging as a promising therapeutic approach. Several RNAi-based clinical trials suggest encouraging results in the treatment of a variety of diseases, including cancer. Here we present miR-Synth, a computational resource for the design of synthetic microRNAs able to target multiple genes in multiple sites. The proposed strategy constitutes a valid alternative to the use of siRNA, allowing the employment of a fewer number of molecules for the inhibition of multiple targets. This may represent a great advantage in designing therapies for diseases caused by crucial cellular pathways altered by multiple dysregulated genes. The system has been successfully validated on two of the most prominent genes associated to lung cancer, c-MET and Epidermal Growth Factor Receptor (EGFR). (See http://microrna.osumc.edu/mir-synth).

**Fig 5.** Examples of (A) a correctly classified article and (B) an incorrectly classified article.

inventory, as a collection of discrete resources, is important to the stakeholders. We explored putting greater weight on precision than $F1$ for both the classification and NER tasks, but we found that the subsequent hit to recall in the NER task resulted in the loss of too many viable predictions. To achieve higher reliability for the inventory then we choose a selectively mediated approach given the relatively small size of the inventory and manageable curation load. Based on the mid-project evaluation results we used the average probability of correctly predicted names, 0.978, to determine a threshold at which resources whose highest scoring predicted name ("best name," regardless if common or full) had a probability below this threshold would be flagged for review by a curator in our finalized process. In the 468 articles from the 10% random sample, records with probabilities $\geq 0.978$ (i.e. those proposed to not undergo human-mediated review in the future) contained 13/468 (2.8%) incorrect classifications and 5/468 (1.0%) incorrect best names. Therefore, we concluded that a selective manual evaluation at this threshold was likely to catch the majority of both classification errors and name errors and improve the quality of the final inventory.

In the mid-project evaluation, 7 articles from the classification test and 5 articles from the NER test set were found in the validation set used for mid-project evaluation. We considered retroactively removing/replacing these 7 articles to address the potentially deleterious effects of such data leakage but decided against these actions for several reasons. First, these articles made up only a very small portion of the mid-project evaluation set, and we did not look at

### Correct Extraction of Common and Partial Extraction of Full Name

**Prediction: Common Name: ESTHER (0.9933) Full Name: Hydrolase (0.7105)**

**PMID:** 23010363 **Title-Abstract:** Proteins with an alpha/beta hydrolase fold: Relationships between subfamilies in an ever-growing superfamily. Alpha/beta hydrolases function as hydrolases, lyases, transferases, hormone precursors or transporters, chaperones or routers of other proteins. The amount of structural and functional available data related to this protein superfamily expands exponentially, as does the number of proteins classified as alpha/beta hydrolases despite poor sequence similarity and lack of experimental data. However the superfamily can be rationally divided according to sequence or structural homologies, leading to subfamilies of proteins with potentially similar functions. Since the discovery of proteins homologous to cholinesterases but devoid of enzymatic activity (e.g., the neuroligins), divergent functions have been ascribed to members of other subfamilies (e.g., lipases, dipeptidylaminopeptidase IV, etc.). To study the potentially moonlighting properties of alpha/beta hydrolases, the **ESTHER** database (for ESTerase and alpha/beta **Hydrolase** Enzymes and Relatives; http://bioweb.ensam.inra.fr/esther), which collects, organizes and disseminates structural and functional information related to alpha/beta hydrolases, has been updated with new tools and the web server interface has been upgraded. A new Overall Table along with a new Tree based on HMM models has been included to tentatively group subfamilies. These tools provide starting points for phylogenetic studies aimed at pinpointing the origin of duplications leading to paralogous genes (e.g., acetylcholinesterase versus butyrylcholinesterase, or neuroligin versus carboxylesterase). Another of our goals is to implement new tools to distinguish catalytically active enzymes from non-catalytic proteins in poorly studied or annotated subfamilies.

**Fig 6. Example of correct extraction of common and partial extraction of full name.** Example of a detailed abstract where the common name of the resource, "ESTHER" (bold/blue), was only mentioned once but correctly predicted. The predicted full name, "Hydrolase" (bold/orange), was partially correct while the entire full name (peach) is much longer. The uncertainty is accurately reflected in the scores where the correctly predicted common name is associated with a probability of 0.9933 while the partially correct full name was lower at 0.7105.

them specifically, so their effects on decisions regarding ML model training/selection were negligible. Second, the only decision made regarding ML training/selection due to the mid-project evaluation was to use precision rather than $F$1-score for article classification model selection. Due to the very low portion of test set articles in the evaluation ($\leq$1.5%), we are confident that such a decision would have been reached in their absence. Since we believe no portion of the model training or selection process was biased by the presence of these articles, the test sets still serve as representative samples for model evaluation. Third, retroactively removing/replacing these 7 articles would result in convoluting an already complex pipeline and reporting of methods. Ultimately, since our goal was a practical application of ML methods, we opted to simply openly disclose the issue.

**3.3.2. Selective manual evaluation of preliminary inventory.** With the results above, the pipeline was implemented and the preliminary inventory up to the point of manual evaluation was created. Of the 3566 records remaining after automated deduplication, 1033 (29.0%) were flagged for review due to a probability score < 0.978, 469 (13.2%) due to potential duplicate names, and 215 (6.0%) due to potential duplicate URLs.

Of the 1033 low probability flags, a curator determined that 805 be retained in the inventory and 228 (6.4% of the total and 22.1% of the flagged) be removed. The majority of those removed, 161/228 (70.6%), were removed because of a partially correct name, while 36/228 (15.8%) were removed for an incorrect name, 27/228 (11.8%) for incorrect classification, and 4/228 (1.8%) for erroneous URLs. As expected, the average probability of those removed (0.749) was much lower than the cut-off that triggered the review.

Of the 469 flagged for duplicate names, 355 (75.7%) were marked for merger, 54 (11.5%) were associated with records that would be removed due to low probability, 50 (10.7%) were not to be merged, and 10 (2.1%) required a partial merger (e.g. sets of > 2 potential duplicates where only some should be merged). Of the 215 flagged for duplicate URLs, 121 (56.3%) were

marked for merger, 66 (30.7%) were associated with records that would be removed due to low probability, and 28 (13.0%) were not to be merged.

## 3.4. URL analysis and testing

In the mid-project manual evaluation we found that, although rare, multi-URL abstracts pose a problem. In the 10% sample, 36/468 (7.7%) of abstracts contained 2 URLs and 4/468 (0.85%) contained > 2 URLs with the remaining containing a single URL. In the majority of 2-URL abstracts (26 of 36, ~ 72%), the correct URL was associated with the predicted name and for 6 of the remaining 10, predictions were low enough to trigger manual review, thereby allowing errors to be caught. However, abstracts with > 2 URLs were more often abstracts that covered multiple distinct biodata resources, such as those that describe a collection of resources located at national data centers. As these abstracts were few in number (representing < 1% of the corpus), they were removed instead of undertaking additional training to associate the correctly predicted name with the correct URL.

In the final inventory 2235/3112 (71.8%) resources had at least one URL which returned HTTP codes in the "2xx successful" or "3xx redirection" series, indicating a live site. However, we note that URLs that resolve with either 2xx or 3xx status codes can be misleading since a URL might resolve successfully but not actually point to the biodata resource. During previous efforts to create a census of databases published in *Nucleic Acids Research*, URLs were manually checked and coded when they failed to resolve to the database in question [74]. As this dataset is available and the websites of interest are the same, we analyzed this dataset to estimate how often URLs that appear to resolve successfully do not, in fact, resolve correctly. The variable "unavailable_message" contained two values for failed 2xx codes ("blank page" and "discontinued notice") and five values for failed 3xx codes ("related generic commercial site redirect", "related generic government site redirect", "related generic publisher site redirect", "related generic research institution site redirect", and "unrelated site redirect"). Of the 2264 URLs that appeared successful in that study, analysis showed that 147 did not resolve properly (6.5%) with 66 resolving to webpages for a related research institution, 37 to an unrelated site, 27 to a discontinued notice, 6 to a related government webpage, 5 to a related publisher webpage, 3 to a related commercial webpage, and 3 to a blank page. We anticipate a similar false positive rate would apply to this work as well. Under this assumption, we estimate that 2090 of the 2235 live URLs pointed to the resource itself, equalling 2090/3112 (67.2%) of the inventory. Previous studies reported ~27% failure while noting that availability is highly time dependent and less popular resources (as determined by citations) are more likely to fail [16]. Additionally, 2451/3112 (78.8%) of resources had at least one URL archived in the Internet Archive's WayBack Machine.

## 3.5. Assessing the biodata resource inventory

After curation and processing to remove erroneous predictions and deduplicate records that were identified in the selective manual evaluation, the resulting inventory contained 3112 resources from 3705 unique articles. Post processing was completed to add additional metadata and check URLs. We provide the following key descriptive statistics to highlight how the inventory may be used to probe for additional information about the resources.

**3.5.1. Countries represented.** Resource locations were assessed by geolocating the URL host IP address and by mining the author affiliations. For 1672 (53.7%) resources at least one IP address could be geolocated, with a total of 1679 IP address locations (Fig 7A). While our methods for identifying countries yielded a consistent output, it misses certain countries, especially for author affiliations where locations are reported more idiosyncratically. For instance,

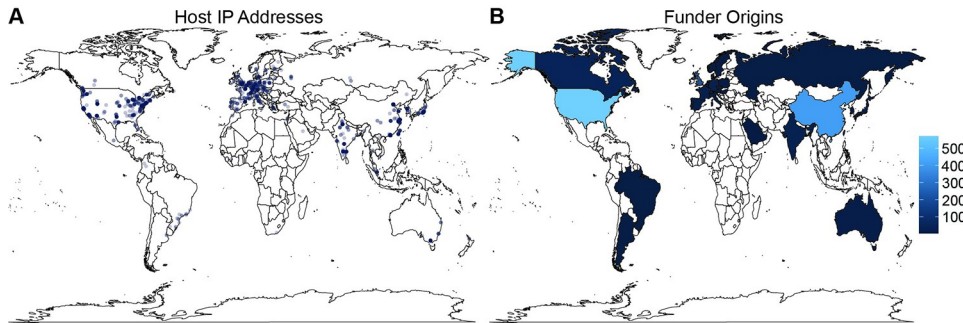

**Fig 7. Biodata resource geolocation metadata.** (A) URL host IP address coordinates (n = 1679). Transparency is constant for all points; darker regions indicate overlapping points. (B) Number of biodata resources per country after determining the country of origin for the most frequently identified funding agencies. Funding agencies that could not be mapped to a single country are not shown; specifically, 111 resources were funded by international agencies. Figures were created using the R ggplot2 package which obtains map data from Natural Earth [71], which is in the public domain.

the ISO-3166-1 name for South Korea is "The Republic of Korea", and if the name does not appear as such in the affiliations, it is missed. Additionally, false positives may occur, such as an affiliation with New Mexico (a state in the USA) being counted as Mexico (the country). However, as a preliminary assessment, 65 unique countries were found in author affiliations (S4 Fig) and 28 unique countries were found in the host IP address locations (S3 Fig). Despite the challenges with cleanly identifying countries, over twice as many countries were identified via author affiliations, indicating that global contributions go well beyond the discrete physical location of a resource. This further highlights the distributed nature of the overall infrastructure.

**3.5.2. Metadata for follow-up analyses of funder and text mining.** Funding of biodata resources can be obtained from the granting agency metadata provided for individual articles. Of the 3705 articles, one or more funding agencies were retrieved for 1916 articles (52.9%), which covers 1714/3112 (55.1%) of the biodata resources identified. Although a total of 1788 "unique" agency names were retrieved, the names reported are variable in that they are often free text values provided by article authors. Both PubMed Central and Europe PMC, which exchange data, make efforts to standardize funder information for their parent funders; however, smaller funders and funders outside of the US and Europe are more likely to be represented irregularly. We see how this challenge is exacerbated in our work by the reporting of increasingly diverse and granular funders as the number of associated biodata resources decreases to only 1 or 2, where reported funder names may be very specific (e.g. an academic department) yet vague (e.g. ambiguous with respect to which university), related to an individual (e.g. scholarships or fellowships of unclear origin), or attributed to a project which, on investigation, is found to be funded itself by multiple funders. Additionally, the funder names retrieved may be of varying organizational levels (e.g. both NIH as a parent and NIH NIGMS as a child), and the same funder can be cited in a single article for distinct grants. In this first instantiation of the inventory itself we left all agency names as is, but future iterations of the inventory may explore use of funder registries, such as the one being developed by CrossRef, to standardize funder names and map identifiers, where possible [72].

To begin assessing the global distribution of the most prevalent funders found here, the funder names associated with 3 or more biodata resources (200/1788, 11.2%) were evaluated by a curator to identify the country or region of the funding organization and verified by a second curator. As noted in prior work, funders themselves are diverse [73], and in addition to

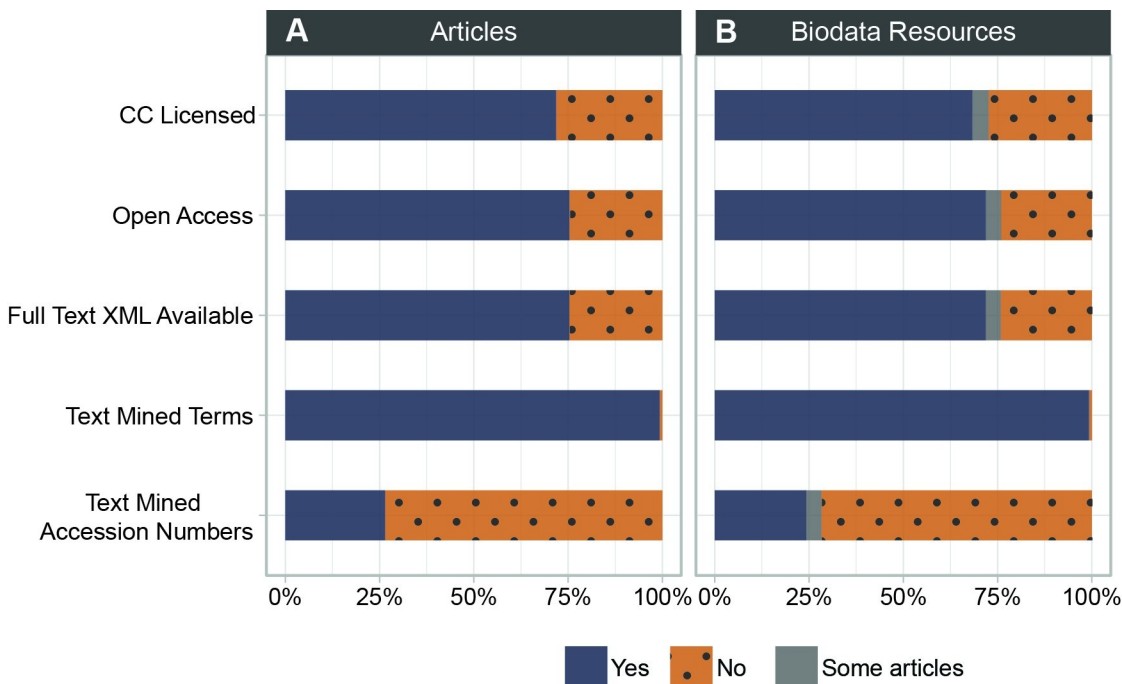

**Fig 8. Inventory text mining potential.** Text mining potential of (A) individual articles found to describe biodata resources, and (B) the collective articles describing a particular biodata resource.

government agencies, academic and philanthropic organizations were also identified here. After deduplication of resources, 28 unique counties were identified, with each supporting anywhere from 3 to 570 biodata resources (Fig 7B).

The articles associated with the biodata resources within the inventory also provide opportunities to glean additional information about the resources themselves through text mining. For each article, Europe PMC provides in-house text mining results as annotations that cover database accessions and resource names of over 60 biodata resources as well as gene/protein names, organisms, diseases, chemicals, gene ontology terms and experimental methods [17]. Additional annotations submitted from the community cover specific interactions, targets, pathways, processes, and other terms [74]. Europe PMC offers a dedicated Annotations API for access to all available terms, and new text mining may also be undertaken on full text available via XML. Thus the vast majority of associated articles have text-mined terms already available for further analyses and the majority are available for additional full text analysis (Fig 8).

**3.5.3. Overlap with resources found in re3data.org and FAIRsharing.** The machine learning-enabled approach used here represents an effort to identify resources at scale. We initially expected greater overlap with re3data and FAIRSharing, but only 536/3112 (17.2%) inventory resources were identified in these two registries; similarly, the majority of life science resources within both re3data (975/1189, 80.5%) and FAIRsharing (1161/1640, 70.8%) are not found in our inventory.

## 3.6. Open science products

From the onset, our goal was to create a fully open project with products that could be reused by anyone and to a high level of reproducibility—we consider this an explicit result of the project and describe these products here briefly (see S7 Table for listing and locations). The final inventory itself, along with all trained models, code, and data are available in both GitHub and

in Zenodo. Within these deposits is the entire workflow, which is provided both as individual scripts as well as an automated Snakemake pipeline that allows execution of the entire analysis in a single command.

In addition to all of the code, iPython notebooks (ipynb) have also been created to facilitate 1) running the entire pipeline with model training and 2) running the prediction script itself from the highest scoring model to update the inventory. Since running the entire pipeline is computationally intensive and requires access to GPUs, the second notebook may be especially useful for updating the inventory without testing or retraining new models. Detailed READ-MEs for all products are available within GitHub and Zenodo, and a dedicated protocol has been created to provide step-by-step instructions for use of the iPython notebooks in Google Colab, which allows anyone to execute the code in a browser.

## 4. Discussion

Collectively, biodata resources form a life sciences infrastructure that is widely distributed and difficult to describe [75]. While there are a number of highly visible and firmly established bio-data resources, others are relatively small and serve specific research communities. In fact, individual resources vary wildly and a persistent, major challenge is simply knowing what bio-data resources exist. Worldwide, several national or regional funders, for example in China, Europe, and the US, support institutes that provide major biodata resources, such as those that host gene sequences and protein structures. Such resources are well-known and easy to locate. However, a biodata resource can be established by anyone who wishes to share data and has access to the skills and technical infrastructure needed to create an online resource; that is, the barrier to entry is relatively low and many resources have been created by individual research-ers motivated to share data. While this increases the availability of data and aligns with global efforts to increase research data sharing, such resources have proliferated to a point where it is no longer possible to know exactly how many resources exist [76]. This creates challenges for anyone searching for data as well as those who work to develop, maintain, and sustain the resources, including resource providers and research funders.

In an effort to address this issue, we implemented a method to identify thousands of biodata resources via a machine learning enabled pipeline. In this practical application of NLP tech-niques we found that several state-of-the-art BERT models performed well in both article clas-sification and NER tasks. The selected article classification model, BioMed-RoBERTa-RCT, achieved very high precision as evaluated on the test set, which provides confidence that the model was not overfit on the training data and that there should be few false positives in the resulting inventory. The low number of false positives is important to maintain inventory qual-ity and also to reduce error propagation since the NER model was used on all articles identified during the classification task. The NER model, also BioMed-RoBERTa-RCT, exhibited lower performance metrics than the article classification model, which may be accounted for in pri-marily two ways. First, determining the name of a resource proved difficult at times even for the curators, especially when the original authors were inconsistent in usage of the resource name. Second, while the article classification task requires a binary classification, the NER task includes five possible classes. This implies that, for instance, if a common name is predicted to be a full name, it is considered a misclassification and has a negative impact on the perfor-mance metrics even though the prediction is still valuable. While this paper presents prelimi-nary efforts to provide a comprehensive proof-of-concept, additional work is already underway to further optimize the ML results. Even as preliminary work, we found these tasks performed well enough that the error rate could have been considered acceptable without remediation. While we ultimately decided to augment with a selective, human-mediated

review to further enhance confidence and veracity in the resulting inventory, we conclude that the application of NLP models is a powerful tool that dramatically aids completion of what would otherwise be an entirely manual, time intensive, and less systematic process. We believe that this work provides a useful approach for addressing a major challenge in sustaining biodata resources—simply being able to monitor an ever-evolving, distributed infrastructure.

The Global Biodata Coalition (GBC) was formed to ensure sustained support for biodata resources and to ensure continuity of the global infrastructure itself. However, the infrastructure itself is not well-defined, thus complicating discussions about sustainability. The GBC initiated this project in order to provide an overview of this infrastructure: how many biodata resources are there, where are they, and how are they funded? The GBC is using the results of the inventory to identify major funders of biodata resources who 1) may be interested in participating in the GBC or 2) are not currently funding biodata resources but might in future. Additionally, research funders themselves have begun using the inventory to identify biodata resources to which they are contributing support.

Previous and current efforts to create collections of data resources tend to rely on individual effort for discovery and curation. The results of such work are high quality and have succeeded in enabling the discoverability of thousands of distinct resources. This inventory does not seek to replace such highly curated collections but potentially it could be used to complement them. The main emphasis of this project was to gather together as many resources as possible using a single, reproducible method. We note, however, that many biodata resources were not identified through our pipeline for several reasons, including 1) biodata resources for which there are no published descriptions, 2) biodata resources described in articles that are not indexed in Europe PMC, and and 3) biodata resources for which descriptive articles have been published but that were missed using our methodology through exclusion, misclassification, or inability to extract a resource name. Furthermore, our corpus only included English-language articles.

While we initially expected to find many of the resources in existing registries, we were surprised by the low overlap. There are several factors that may impact this result. For example, the selection criteria may differ in subtle ways given that there is no universally shared definition of "data" itself. Additionally, some matches may have been missed due to name/URL variation, migrations, or mergers. A comprehensive follow-up that carefully reviews each resource would enable a better understanding of the similarities and differences between the distinct collections. As this follow-up would require a substantial amount of human-mediated curation, other notable collections that were not amenable to the automation utilized here, such as *NAR's* Molecular Biology Database Collection or Database Commons, could be examined as well. One potential use of the inventory is as a catalyst for outreach. For example, journals found to frequently publish biodata resource articles could be approached to encourage or even require registration for those publications. Additionally, the corresponding authors of the resources identified in the inventory could be contacted to encourage registration to increase the discoverability of their resources. While we designed this project assuming that updating the inventory will be necessary, in our ideal future state, all biodata resources are easily located elsewhere.

At the onset of this project, we identified openness and reproducibility as key to enabling updates, reuse, and extension of this work. A preliminary analysis of the inventory indicates that the availability of article metadata and the high percentage of full text articles will indeed enable reuse. In spite of inherent limitations found in the irregular representation of funding organizations and locations in the associated metadata, the inventory already provides a useful glimpse into this difficult-to-describe distributed infrastructure. For example, even a casual scan of funders revealed national organizations from around the world, including but not limited to, the Research Council of Norway, the Spanish Ministry of Science and Innovation, the

Czech Science Foundation, the South African National Research Foundation, the Israel Science Foundation, the Qatar National Research Fund, the Indian Department of Health Research, the National Research Foundation of Korea, the Ministry of Science of Technology of Taiwan, the Australian Research Council, the National Agency for the Promotion of Research, Technological Development and Innovation of Argentina, the Mexican National Council of Science and Technology, and the Oneida Nation Foundation. We hope our efforts to make the inventory completely open and to develop the code in ways that make it accessible to the widest possible audience will help catalyze future work in understanding the global infrastructure of biodata resources.

## Supporting information

**S1 Fig. Europe PMC query.**
(TIF)

**S2 Fig. Example of resource name extraction from combined title and abstract.** Process shows how the tokens are labeled using the BIO scheme and probability scores are output by the linear token classification layer of the BERT model. Tokens are then reassembled into words using the associated word indices (not shown), and the average probability score of the tokens is calculated. Trailing punctuation is removed from predicted resource names.
(TIF)

**S3 Fig. URL host IP address countries.** Choropleth map shows URL host IP address countries based on matches to ISO-3166-1 country names or Alpha-3 codes. Color is scaled to the number of times that country's name appeared as a host IP address location. Figure was created using the R ggplot2 package which obtains map data from Natural Earth [71], which is in the public domain.
(TIF)

**S4 Fig. Author affiliation countries.** Choropleth map shows author affiliation countries based on matches to ISO-3166-1 country names or Alpha-3 codes. Color is scaled to the number of times that country's name appeared in the author affiliations across all articles in the inventory. Figure was created using the R ggplot2 package which obtains map data from Natural Earth [71], which is in the public domain.
(TIF)

**S1 Table. Definitions consulted for "Life sciences biodata".**
(PDF)

**S2 Table. Definitions consulted for "Biodata resource".**
(PDF)

**S3 Table. APIs used.**
(PDF)

**S4 Table. Hyperparameters used for model fine-tuning for article classification and NER tasks.**
(PDF)

**S5 Table. Article classification model performance.** Performance metrics are shown for both the validation and test sets. Models are arranged in decreasing order of precision on the validation set, which was used for model selection.
(PDF)

**S6 Table. NER model performance.** Performance metrics are shown for both the validation and test sets. Models are arranged in decreasing order of *F*1 score on the validation set, which was used for model selection.
(PDF)

**S7 Table. Open science products.**
(PDF)

## Acknowledgments

The authors would like to thank colleagues at the Chan Zuckerberg Initiative, in particular Dario Taraborelli, Donghui Li, Gully Burns, and Emanuele Bezzi, for their support and feedback on earlier versions of this study. We also thank Ken Youens-Clark formerly at The University of Arizona, Alise Ponsero at The University of Helsinki, and Bonnie Hurwitz at The University of Arizona for their mentorship of Kenneth Schackart. Additionally, we thank the Europe PMC team, especially Aravind Venkatesan, Mohamed Selim, and Melissa Harrison, for their guidance and expertise. Finally, we would like to acknowledge Jodie Forbes for detailed review of the associated code and documentation.

To reflect the contributions of the individual authors more fully, the following details are provided in addition to those mapped to the CRediT Taxonomy: HJI—Project conceptualization and planning; Manual data curation; Development of preliminary code for data analysis; Project oversight and administration; Data validation; Writing of manuscript; KES—Manual data curation; Implementation of machine learning methodology; Design and implementation of code for processing and augmenting predicted resources, automation of pipelines, unit testing and static code checks, and data analysis; Creation of data visualizations and figures; Data validation; Writing of manuscript; AMI—Design of the machine learning methodology; Implementation of code for training, prediction and evaluation of the NLP models used to classify articles and extract individual resources; Writing of manuscript; CEC—Project conceptualization and planning; Manual data curation; Funding acquisition; Project oversight; Data validation; Writing of manuscript.

## Author Contributions

**Conceptualization:** Heidi J. Imker, Kenneth E. Schackart, III, Ana-Maria Istrate, Charles E. Cook.

**Data curation:** Heidi J. Imker, Kenneth E. Schackart, III, Charles E. Cook.

**Formal analysis:** Heidi J. Imker, Kenneth E. Schackart, III, Ana-Maria Istrate.

**Funding acquisition:** Charles E. Cook.

**Investigation:** Heidi J. Imker, Kenneth E. Schackart, III, Ana-Maria Istrate.

**Methodology:** Heidi J. Imker, Kenneth E. Schackart, III, Ana-Maria Istrate.

**Project administration:** Heidi J. Imker.

**Software:** Kenneth E. Schackart, III, Ana-Maria Istrate.

**Supervision:** Charles E. Cook.

**Validation:** Heidi J. Imker, Kenneth E. Schackart, III, Ana-Maria Istrate.

**Visualization:** Kenneth E. Schackart, III, Ana-Maria Istrate.

**Writing – original draft:** Heidi J. Imker, Kenneth E. Schackart, III, Ana-Maria Istrate, Charles E. Cook.

**Writing – review & editing:** Heidi J. Imker, Kenneth E. Schackart, III, Ana-Maria Istrate, Charles E. Cook.

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
