## [Decision Letter · Decision Letter 0]

31 Jul 2023

PONE-D-23-16263A machine learning-enabled open biodata resource inventory from the scientific literaturePLOS ONE

Dear Dr. Imker,

Thank you for submitting your manuscript to PLOS ONE. After careful consideration, we feel that it has merit but does not fully meet PLOS ONE’s publication criteria as it currently stands. Therefore, we invite you to submit a revised version of the manuscript that addresses the points raised during the review process.

We look forward to receiving your revised manuscript.

Kind regards,

Anna Bernasconi, PhD

Academic Editor

PLOS ONE

“This work was supported by the Chan Zuckerberg Initiative (chanzuckerberg.com), which is a member of the Global Biodata Coalition. This work was also funded by the Global Biodata Coalition (globalbiodata.org), a coalition of research funding organizations working towards sustainability of biodata resources worldwide.”

3. We note that Figures 7,S3 and S4 in your submission contain [map/satellite] images which may be copyrighted. All PLOS content is published under the Creative Commons Attribution License (CC BY 4.0), which means that the manuscript, images, and Supporting Information files will be freely available online, and any third party is permitted to access, download, copy, distribute, and use these materials in any way, even commercially, with proper attribution. For these reasons, we cannot publish previously copyrighted maps or satellite images created using proprietary data, such as Google software (Google Maps, Street View, and Earth). For more information, see our copyright guidelines: http://journals.plos.org/plosone/s/licenses-and-copyright.

a. You may seek permission from the original copyright holder of Figures 7,S3 and S4 to publish the content specifically under the CC BY 4.0 license. 

Additional Editor Comments:

Dear authors, please take into account all the comments made by the reviewers, who have thoroughly examined your manuscript. Please provide a revised version where changes/additions are highlighted in different color. Especially Rev. 2, who largely appreciated your work, has many suggestions for improvement and for increasing its readership. We will be glad to re-evaluate your work after.

Reviewers' comments:

Reviewer's Responses to Questions

**Comments to the Author**

1. Is the manuscript technically sound, and do the data support the conclusions?

Reviewer #1: Yes

Reviewer #2: Partly

Reviewer #3: Yes

2. Has the statistical analysis been performed appropriately and rigorously? 

Reviewer #1: Yes

Reviewer #2: N/A

Reviewer #3: Yes

3. Have the authors made all data underlying the findings in their manuscript fully available?

Reviewer #1: Yes

Reviewer #2: Yes

Reviewer #3: Yes

4. Is the manuscript presented in an intelligible fashion and written in standard English?

Reviewer #1: Yes

Reviewer #2: Yes

Reviewer #3: Yes

5. Review Comments to the Author

Reviewer #1: General appreciation: The ML pipelines for the identification of biodata resources were diligently constructed with several verifications along the road. The results have been validated in several manners. The publication is mostly about the method, not about the resulting inventory (which may be at least as interesting).

Error to be corrected: Line 489: the figure says ESTHER, not ESTER

Elements that would be valuable to amend are:

* Lines 672-673: describes the "WHY?" of this inventory. However, this is not well explained nor convincing. The Global Biodata Coalition is about sustaining important databases worldwide and this publication aims to establish a method to describe the background landscape of data resources. Why is this useful? Why do we need the inventory and/or what will be done with it? How does it feed into the work of the Coalition?

* There is no mentioning about the language of the underlying data resources (this is on a corpus of English-language publications). This seems nevertheless relevant for a global inventory.

* Section 3.5.3: The section on the comparison with re3data.org and FAIRSHARING could be worked out more diligently. The overlap is less than 20%, leading the authors to state that both methods are complementary. Being complementary implies that their purpose is similar and that the 3 inventories will feed into the work of the Global Biodata Coalition. (This again points to the WHY? of the work as they can only be complementary if the aim is similar.) There are alternative explanations possible that are not addressed: The underlying collections of resources are not at all comparable and therefore both inventories are not valid benchmarks. Or the method described by the authors is missing a large part of the resources. Further clarification is therefore needed. NAR would have been a useful source for a verification. Is there any other way to use the NAR data for verification through a human, manual check as the FAIRSHARING/re3data.org verification has not allowed any firm conclusions?

* Section 3.4 about URL testing: 29% of the URLs did not resolve. There would be another 6.5% false positives. We know that many data resources are archived over time or disappear. Is this number of about 35% of data resources unavailable within what is expected based on the literature?

Reviewer #2: Overall, the authors make a compelling case for the need for a pipeline to find biodata resources automatically based on the reliance of biological science on such resources. The focus on open science is also very valuable for future use and updates of such a repository of biodata resources. At the same time, the article needs more details on specific design choices as well as more exploration of the resulting repository to provide evidence for the authors' claims. Further, from a writing perspective, some of the sentences are too long and some content needs to be reorganized into different article sections. In general though, the idea is interesting and warrants publication after revisions. See below for more detailed revisions (in order of the article).

Major Revisions:

1. There are many long sentences. Example, in the introduction the last sentence of the 2nd paragraph is very long and confusing ("However, and again unlike physical sciences,..."). Another example is the last sentence of the first paragraph in the methods ("while there are limitations...").

2. One of the discussion points in the end is that the list of data repositories found can be used to contact the owners of the data repositories to register them. This point though is buried in the introduction in the paragraph that starts "Blair et al". More details about registering databases is necessary to foreshadow this finding and the importance of it in general.

3. The query used to retrieve publications that contain a data repository is not well explained. It is mentioned that they "developed a targeted query", but it is not clear to me how that was developed. More details appear later about the issue with other types of URLs. While later Figure S1 is pointed to with the query, it is not explained (data resources page 9). In terms of the query itself, the term "repository" is notably missing. Also "knowledge-base". Further, for training purposes on the article classifier, it is interesting that there are no true negative articles in the dataset. Meaning random articles that do not talk about data resources at all. On some level, one could think of any new article and classifying it as having a data resource or not immediately without the targeted query each time. Also, new words for data resources may appear and so would the query be updated to accommodate for this? In general, more details on the corpus construction would be beneficial.

4. One of the main evaluations of this work is a comparison to other data inventory organizations like re3data and FAIRsharing. In the methods, there needs to be an acknowledgement of the different possibilities with this comparison (i.e., what does a large overlap mean vs. a small overlap) and that in general, these inventories do not include all repositories. This should foreshadow section 3.5.3.

5. There needs to be a clear distinction between the methods and results sections. For example, under section 2.5.1, the last few sentences of the first paragraph are results. The methods should state that inter-annotator agreement was calculated and how it was calculated and that an error analysis was conducted. The results should be in results. It is not clear to me also where discussion fits into this. I think a separate section would be helpful. Without this distinction there is a fair amount of repetitive content.

6. A discussion on the interplay between the two tasks (article classification and NER) needs to be discussed. Errors will propagate between tasks.

7. It is mentioned that the best algorithm is chosen based on the validation set and I am unclear why this is not on the test set? This needs to be further clarified.

8. Precision was chosen over F1 for article classification. This is somewhat explained later and clearly recall is low based on the results. However, in methods this needs to be explained. Ideally, the motivation for focusing on different metrics exists in the introduction. (page 15) Some explanation exists on page 25 with "we explored putting greater weight on precision than F1…".

9. Section 2.9 needs more details on the analyses performed on the inventory itself. The results mention a lot of methods around this that should be moved here. Further, other types of analyses like the publication dates of repositories, the number of articles each repository has, etc. would continue to highlight the value of this work.

10. The NLP results are interesting (section 3.2) and need to be discussed more. The precision is very high for the article classification and it is not clear to me why that is true. Also the test and validation sets disagree on the best performing algorithm. The test set has very high scores also. A mention of this here would be good to foreshadow the issue in data that is mentioned later.

11. The URL information should be introduced earlier (section 3.4) with the background on the analysis of resolving URLs and issues previous research has had. A discussion section is missing and so it is confusing that results also have discussion in it. Sections 3.5.1 and 3.5.2 are much more discussion based then just results. In 3.5.2, future work is mentioned that should be in a discussion section.

12. The overlap comparison (section 3.5.3) needs more details in it and a further analysis. The authors speculate at the differences but if the authors want to claim that their work complements the other work, then it needs to be clear how. Should all the repositories the authors found be added into the other inventories? Do they have different purposes? This is especially crucial when they claim they could reach out to authors to add their databases.

Minor Revisions:

1. A citation is missing for Blair et al on page 5.

2. Does the working definition of biodata include the data in biomedical knowledge-bases or ontologies?

3. It is mentioned that models are trained for max of 10 epochs. Why?

4. Classically the BIO/IBO/IOB scheme is B = beginning, I = inside, and O = outside. The otherwise is interesting here (page 15)

5. The mid-project evaluation is interesting and creates a very high probability threshold. The exact threshold is more results and so more explanation of where the threshold came from would be beneficial.

6. It is not clear how much work the manual curators did for the manual annotation. (The time they spent is also results.) Since the motivation is automation and saving time, more information on the time would be beneficial.

7. Please add percentages with the fractions. Also put the resulting numbers in parentheses without explaining it is F1 score). This will get rid of a lot of unnecessary text.

8. Figures:

a. Color does not show up with black and white printing. Check how it looks in black and white.

b. Figure 1 is blurry.

c. Figure 3 is a little confusing going bottom to top. I think it would be less confusing flipping it from top to bottom. It is also not clear what classification layer was used for the algorithms.

d. Figure 7 and 8 are hard to read in black and white printing.

9. How many articles were reviewed in the 10% sample (page 25). It is hard to understand with the denominator of 468. It seems that 18 were reviewed (13+5).

Exceptional:

1. The open science implementation plan is amazing! I like that the authors are implementing what they believe in.

2. It is great that the authors deduplicate articles but still save all articles. Maybe both the original article and the most recent one could be saved?!

3. Thank you for disclosing the data issues faced.

Reviewer #3: The authors of the submitted manuscript report on a thorough work on producing an open catalogue of biodata resources. They describe the process of collecting and verifying information for such a catalogue. The authors used ML, NLP plus manual adjustments. The code and the catalogue itself are made available to the research community. While the used methods are straight forward, their usage is well justified and applying them to a large corpus of scientific publication is a time-consuming process. The manuscript is well written and structured, the reported results are of great importance. Therefore, I believe this manuscript should be accepted for a publication after minor revisions.

Suggestions for improvement of the current draft:

[133 ‘used a machine-learning based approach’ It is not clear at the beginning of the manuscript what ml was employed? Is it using BERT? It should be explained from the beginning.

In my view, the manuscript could be shortened. For example, there is a bit of repetition in sections 2 and 3. Also, some parts could be moved to SI (supplementary information).

[555] ‘111 resources were funded by international agencies...’ It is a lot of resources. It is not clear why funding agencies that could not be mapped to a single country were not shown.

[623] ‘the majority of life science resources within both re3data (975/1189, 80.5%) and FAIRsharing 635 (1161/1640, 70.8%) are not found in our inventory’. That is a real worry. This warrants a more detailed investigation, explanation and suggestion for a resolution.

Personally, I do not like the title: ‘A machine learning-enabled open biodata resource inventory from the scientific literature’. I suggest or to be more specific what ml was used, or drop mentioning of ml. I do not see why it is so important that ml was used to make it to the title. The important aspects are: open, re-usable, updatable, scientific literature driven.

Just to add, it was interesting to see the results of Mid-Project evaluation and iteration.

6. PLOS authors have the option to publish the peer review history of their article (what does this mean?). If published, this will include your full peer review and any attached files.

Reviewer #1: No

Reviewer #2: No

Reviewer #3: **Yes: **Larisa Soldatova

---

## [Author Response · Author response to Decision Letter 0]

14 Sep 2023

Our responses are provided in the uploaded Cover Letter and Response to Reviewers files. We will paste that text below, but it is probably much more readable in the PDFs provided as we do not know if line breaks will be preserved here. 

EDITOR/Journal Requirements

** Our Response: Our resubmission has been checked against the PLOS ONE’s style requirements including file naming. We believe it is compliant. 

“This work was supported by the Chan Zuckerberg Initiative (chanzuckerberg.com), which is a member of the Global Biodata Coalition. This work was also funded by the Global Biodata Coalition (globalbiodata.org), a coalition of research funding organizations working towards sustainability of biodata resources worldwide.”

Please state what role the funders took in the study.

** Our Response: Thank you for applying the changes online on our behalf. Our amended statement is as follows:

“This project was initiated by the Global Biodata Coalition as part of its programme of work, and which supported the work of CEC, KES, and HJI in planning and implementing the project. The Chan Zuckerberg Initiative supported the work of AMI in the development of machine learning methods.”

3. We note that Figures 7,S3 and S4 in your submission contain [map/satellite] images which may be copyrighted.

** Our Response: Figures 7, S3, and S4 were created using the R package ggplot2 (specifically the function ggplot2::map_data()), which gets the data using the R package maps. The documentation for the maps package states that the database used is the Natural Earth data project which, as listed above, is in the public domain (see https://cran.r-project.org/web/packages/maps/readme/README.html). We have added this information to the figure captions and cited Natural Earth as necessary.

Additionally, all figures have been checked per reviewer 2’s request and they have been checked in PACE per the decision letter request. The figure files provided in the revision have been downloaded from PACE. 

REVIEWER Prompts/Comments

1. Is the manuscript technically sound, and do the data support the conclusions?

Reviewer #1: Yes

Reviewer #2: Partly

Reviewer #3: Yes

** Our Response: Actions taken to address Reviewer 2’s “Partly” are below. 

2. Has the statistical analysis been performed appropriately and rigorously?

Reviewer #1: Yes

Reviewer #2: N/A

Reviewer #3: Yes

** Our Response: No action needed. 

3. Have the authors made all data underlying the findings in their manuscript fully available?

Reviewer #1: Yes

Reviewer #2: Yes

Reviewer #3: Yes

** Our Response: No action needed. 

4. Is the manuscript presented in an intelligible fashion and written in standard English?

Reviewer #1: Yes

Reviewer #2: Yes

Reviewer #3: Yes

** Our Response: No action needed. 

5. Review Comments to the Author

Reviewer #1: 

General appreciation: The ML pipelines for the identification of biodata resources were diligently constructed with several verifications along the road. The results have been validated in several manners. The publication is mostly about the method, not about the resulting inventory (which may be at least as interesting).

** Our Response: We are gratified to read these positive comments. We did choose to focus this publication on the method itself, but we agree that the inventory is just as interesting and have commented below on potential follow-up work.

Error to be corrected: Line 489: the figure says ESTHER, not ESTER

** Our Response: This error has been corrected. Thank you for the keen eyes. 

Elements that would be valuable to amend are:

Lines 672-673: describes the "WHY?" of this inventory. However, this is not well explained nor convincing. The Global Biodata Coalition is about sustaining important databases worldwide and this publication aims to establish a method to describe the background landscape of data resources. Why is this useful? Why do we need the inventory and/or what will be done with it? How does it feed into the work of the Coalition?

** Our Response: We appreciate this feedback and have added a paragraph to make the purpose and value of this work more explicit in Section 4. Discussion.

There is no mentioning about the language of the underlying data resources (this is on a corpus of English-language publications). This seems nevertheless relevant for a global inventory.

** Our Response: Thank you for catching this omission as this is very important to note. We edited the methods section to state that the Europe PMC API was accessed to obtain English-language articles, and we further revised to include this point in the discussion as well. 

Section 3.5.3: The section on the comparison with re3data.org and FAIRSHARING could be worked out more diligently. The overlap is less than 20%, leading the authors to state that both methods are complementary. Being complementary implies that their purpose is similar and that the 3 inventories will feed into the work of the Global Biodata Coalition. (This again points to the WHY? of the work as they can only be complementary if the aim is similar.) There are alternative explanations possible that are not addressed: The underlying collections of resources are not at all comparable and therefore both inventories are not valid benchmarks. Or the method described by the authors is missing a large part of the resources. Further clarification is therefore needed. NAR would have been a useful source for a verification. Is there any other way to use the NAR data for verification through a human, manual check as the FAIRSHARING/re3data.org verification has not allowed any firm conclusions?

** Our Response: These are all terrific points, and we revised the text in the discussion to clarify. As the reviewer points out, there are many possible explanations. From the preliminary work presented, we know that carefully teasing apart the possibilities would certainly require a substantial amount of human curation and would constitute another study entirely. Our goal with this publication was to gauge reception of and interest in the inventory and our methods for assembling it. So while a more detailed analysis is out of the scope of this publication, which is already quite lengthy and detailed (too lengthy according to another reviewer), we absolutely agree that it is warranted and are very happy to hear there is interest in additional analyses.

Section 3.4 about URL testing: 29% of the URLs did not resolve. There would be another 6.5% false positives. We know that many data resources are archived over time or disappear. Is this number of about 35% of data resources unavailable within what is expected based on the literature?

** Our Response: It’s roughly within expectations, although on the high side when adding in the false positive estimate. Previous studies reported ~27%, noting that it’s highly time dependent and also appears related to popularity (as determined by citations). We haven’t assessed the up/down breakdown by year or citation, but those values are included in the inventory if anyone wants to investigate. We added text to this extent in this section.

Reviewer #2: 

Overall, the authors make a compelling case for the need for a pipeline to find biodata resources automatically based on the reliance of biological science on such resources. The focus on open science is also very valuable for future use and updates of such a repository of biodata resources. At the same time, the article needs more details on specific design choices as well as more exploration of the resulting repository to provide evidence for the authors' claims. Further, from a writing perspective, some of the sentences are too long and some content needs to be reorganized into different article sections. In general though, the idea is interesting and warrants publication after revisions. See below for more detailed revisions (in order of the article).

** Our Response: In addition to the specific response below, we’d sincerely thank this reviewer for their careful reading and thoughtful consideration of our work. It is gratifying to see appreciation for our efforts toward open science and reproducibility. Their detailed suggestions and comments have been very helpful in shaping this manuscript into a more readable and useful addition to the literature. The reviewer did have many suggestions for restructuring that we considered in all cases and fully or partially implemented in many. We are very sympathetic to the reviewer’s preference for a crisper and more linear presentation. However, we found this paper particularly difficult to write, in part because of the interactive nature of the process and the mid-project evaluation. Because of this, we deliberately sought out and followed PLOS One’s guidance for structuring a discussion as we prepared the manuscript. This guidance (https://plos.org/resource/how-to-write-conclusions/) encourages authors to focus on key findings. We have reworked the paper considerably in light of the reviewer’s comments, but we did not revise the discussion to systematically cover all findings as the reviewer seemed to request. We also note that the other two reviewers did not seem to mind the structure we presented, with one specifically stating they found the manuscript well written and structured.

Major Revisions:

1. There are many long sentences. Example, in the introduction the last sentence of the 2nd paragraph is very long and confusing ("However, and again unlike physical sciences,..."). Another example is the last sentence of the first paragraph in the methods ("while there are limitations...").

** Our Response: We have made edits to reduce long sentences and improve the readability of our manuscript throughout, including the two examples given.

2. One of the discussion points in the end is that the list of data repositories found can be used to contact the owners of the data repositories to register them. This point though is buried in the introduction in the paragraph that starts "Blair et al". More details about registering databases is necessary to foreshadow this finding and the importance of it in general.

** Our Response: This is a very helpful suggestion.Text has been added. 

3. The query used to retrieve publications that contain a data repository is not well explained. It is mentioned that they "developed a targeted query", but it is not clear to me how that was developed. More details appear later about the issue with other types of URLs. While later Figure S1 is pointed to with the query, it is not explained (data resources page 9). In terms of the query itself, the term "repository" is notably missing. Also "knowledge-base". Further, for training purposes on the article classifier, it is interesting that there are no true negative articles in the dataset. Meaning random articles that do not talk about data resources at all. On some level, one could think of any new article and classifying it as having a data resource or not immediately without the targeted query each time. Also, new words for data resources may appear and so would the query be updated to accommodate for this? In general, more details on the corpus construction would be beneficial.

** Our Response: We’ve added text on query development to the method section. We’ve also used “repository” as an example term which is used in different, but related, contexts that creates other challenges (e.g. conflation with code repositories and especially those associated with data analysis servers). We explicitly designed the pipeline so that a user-provided query can be provided. We’re happy to make that feature more explicit, so this text was added there, too. 

Regarding the “no true negatives” we see this point in theory, but in practice it did not work that way entirely. Although we developed a query to enrich the set with articles describing a resource, from our curation efforts we know that it didn’t entirely exclude those that didn’t. These were not always random articles but still true negatives for our purposes. For example, articles that described use of data and then referenced the URL for that data source in the abstract were common. Some others were quite left field. For example, computer science articles that included “WWW” and “data” in the abstract. To train the classifier on all articles certainly could be an interesting follow-up effort but a substantial one both in terms of labeling data and computation. For this effort, what we hoped to learn was if this strategy was going to help us generate a reasonably accurate inventory and what all would be needed post-NLP to make that inventory of use to those who we believed would be interested in it. 

4. One of the main evaluations of this work is a comparison to other data inventory organizations like re3data and FAIRsharing. In the methods, there needs to be an acknowledgement of the different possibilities with this comparison (i.e., what does a large overlap mean vs. a small overlap) and that in general, these inventories do not include all repositories. This should foreshadow section 3.5.3.

** Our Response: This is a good point that may not be obvious to everyone. We have added text to address this observation in the methods, which now also contains additional details suggested in other points made by the reviewer. 

5. There needs to be a clear distinction between the methods and results sections. For example, under section 2.5.1, the last few sentences of the first paragraph are results. The methods should state that inter-annotator agreement was calculated and how it was calculated and that an error analysis was conducted. The results should be in results. It is not clear to me also where discussion fits into this. I think a separate section would be helpful. Without this distinction there is a fair amount of repetitive content.

** Our Response: We are deeply sympathetic to this comment. When we tried a stricter delineation, the methods seemed disjointed without some allusion to what was learned and why we did x, y, z. We have taken a fresh look throughout though and pointed readers to results instead in more places, for example in the IAA. We have also revised to remove redundant content wherever possible. Please see our opening response regarding the discussion section. 

6. A discussion on the interplay between the two tasks (article classification and NER) needs to be discussed. Errors will propagate between tasks.

** Our Response: This is a good point, and we revised the text in the discussion to touch on this. In particular, we find that since the article classification model had such high precision, the risk of error propagation is minimized, but it is certainly true that the names of things which are not biodata resources may still be predicted by the NER model.

7. It is mentioned that the best algorithm is chosen based on the validation set and I am unclear why this is not on the test set? This needs to be further clarified.

** Our Response: We are following the convention for ML model selection as can be found in Wulf et al., 2022 (now cited in the manuscript) which seemed most appropriate to our use case. We have revised the text in the methods section 2.5.1 when introducing the training data sets.

8. Precision was chosen over F1 for article classification. This is somewhat explained later and clearly recall is low based on the results. However, in methods this needs to be explained. Ideally, the motivation for focusing on different metrics exists in the introduction. (page 15) Some explanation exists on page 25 with "we explored putting greater weight on precision than F1…".

** Our Response: We’ve added text to the introduction and in Section 2.5.3. for the article classification task. 

9. Section 2.9 needs more details on the analyses performed on the inventory itself. The results mention a lot of methods around this that should be moved here. Further, other types of analyses like the publication dates of repositories, the number of articles each repository has, etc. would continue to highlight the value of this work.

** Our Response: We’ve added more detail, rearranged the results and methods some, and also moved supplemental material into newly created subsections in Section 2.9. With regard to additional analyses, our goal with this publication was to gauge reception of and interest in the inventory and our methods for assembling it. The preliminary analyses were intended to highlight what could be done with the inventory since there are many, many avenues for further analyses. In fact, the ideas suggested here are one direction and another reviewer suggested an entirely different direction. This article is already quite lengthy and complicated (another reviewer suggested we shorten it), so we believe more detailed analyses are out of the scope for this publication. However, we absolutely agree that it is warranted and are very happy to hear there is interest in additional analyses. 

10. The NLP results are interesting (section 3.2) and need to be discussed more. The precision is very high for the article classification and it is not clear to me why that is true. Also the test and validation sets disagree on the best performing algorithm. The test set has very high scores also. A mention of this here would be good to foreshadow the issue in data that is mentioned later.

** Our Response: We have added more text to the discussion section to address the high precision of the article classification model on the test set, the lower performance of the NER model, and potential explanations for differences in performance. While it is true that the test and validation sets disagree, the model is selected based on the validation set, otherwise researchers risk biasing their selection to the test set results, rendering them no longer a good indicator of how the model will perform on unseen data.

11. The URL information should be introduced earlier (section 3.4) with the background on the analysis of resolving URLs and issues previous research has had. A discussion section is missing and so it is confusing that results also have discussion in it. Sections 3.5.1 and 3.5.2 are much more discussion based then just results. In 3.5.2, future work is mentioned that should be in a discussion section.

** Our Response: We’ve moved text out of supplements into Section 3.4 to try address the reviewer’s comments, but it is not clear to us where we would have introduced the URL information earlier so we respectfully decline this suggestion. Please see the opening response above in regards to the discussion. 

12. The overlap comparison (section 3.5.3) needs more details in it and a further analysis. The authors speculate at the differences but if the authors want to claim that their work complements the other work, then it needs to be clear how. Should all the repositories the authors found be added into the other inventories? Do they have different purposes? This is especially crucial when they claim they could reach out to authors to add their databases.

** Our Response: Details have been added to the methods section to describe the registries, their selection criteria, and our access, filtering, and cleaning processes. This was an oversight on our part not to include this information, and we appreciate that the reviewer caught this omission. Additionally, we revised the discussion to state more clearly how our work may complement existing registries without saying that it does already and to emphasize that additional work must be done. To the best of our knowledge, neither re3data or FAIRsharing’s processes for identifying resources are assisted by ML, so it seems to us that the method itself is one point of potential complementarity. It is not clear yet the extent to which the inventory will complement the actual collections of these registries, so we see how this may have been too premature to state. This has been revised as well. As far as additional analyses to get at that, we know from the preliminary work presented that this will certainly require a substantial amount of human curation and would constitute another study entirely. Similarly, “Should all the repositories the authors found be added into the other inventories?” is a terrific question, but it’s not a question we can answer in the present study nor do we think we should try to address it alone. We’re simply noting that since we have corresponding author information, at least such outreach is now a possibility. 

Minor Revisions:

1. A citation is missing for Blair et al on page 5.

** Our Response: The citation has been added. Thank you for the keen eyes.

2. Does the working definition of biodata include the data in biomedical knowledge-bases or ontologies?

** Our Response: Computer Retrieval of Information on Scientific Projects Thesaurus (CRISP), National Cancer Institute Thesaurus (NCIT), Data Catalog Vocabulary (DCAT), and the Biomedical Resource Ontology (BRO) were all consulted as we put together definitions (see Tables S1 and S2). Or is the reviewer asking if these would meet the criteria for a biodata resource in our study? If so, yes, although we did not check to see if any of these surfaced. 

3. It is mentioned that models are trained for max of 10 epochs. Why?

** Our Response: This was based on general convention as a starting point. The text has been revised to state this.

4. Classically the BIO/IBO/IOB scheme is B = beginning, I = inside, and O = outside. The otherwise is interesting here (page 15)

** Our Response: This was a typo that we've corrected. Thank you for the careful reading. 

5. The mid-project evaluation is interesting and creates a very high probability threshold. The exact threshold is more results and so more explanation of where the threshold came from would be beneficial.

** Our Response: This comment seems to make our point above about the challenge between the methods and results section for this article. Is it “more results”? When we did that originally, it seemed awkward that it wasn’t in the methods. We currently have an explicit parenthetical to reference readers to the results in that methods section, so we believe it’s best to leave it as is. 

6. It is not clear how much work the manual curators did for the manual annotation. (The time they spent is also results.) Since the motivation is automation and saving time, more information on the time would be beneficial.

** Our Response: We have this documentation so we revised the text in the methods to include the hours spent on curating the training set. We appreciate the reviewer’s attention to such details throughout.

7. Please add percentages with the fractions. Also put the resulting numbers in parentheses without explaining it is F1 score). This will get rid of a lot of unnecessary text.

** Our Response: We have revised to add percentages and reviewed the text throughout. Note that because we chose to use precision in for classification, we were deliberately explicit in which metric was used in an effort to keep it clear for readers.

8. Figures:

a. Color does not show up with black and white printing. Check how it looks in black and white.

b. Figure 1 is blurry.

c. Figure 3 is a little confusing going bottom to top. I think it would be less confusing flipping it from top to bottom. It is also not clear what classification layer was used for the algorithms.

d. Figure 7 and 8 are hard to read in black and white printing.

** Our Response: Thank you for pointing out the issues with our figures. Figure 1 has been re-exported to ensure image quality. For Figure 3, the bottom-to-top orientation isn’t unusual to see (e.g., Devlin et al., 2019 https://doi.org/10.48550/arXiv.1810.04805), and it’s our own preference so we prefer to leave it as is. Regarding the classification layer, the text is more explicit by stating that a linear classification layer is used. We believe this figure borders on too complex already, so we prefer to leave it as is. Figure 7 has been modified such that the land-masses are white and the borders are drawn in black. We believe this improves contrast and readability in black and white. We have added a patterned fill to figure 8 to improve readability in black and white.

9. How many articles were reviewed in the 10% sample (page 25). It is hard to understand with the denominator of 468. It seems that 18 were reviewed (13+5).

** Our Response: All 468 were reviewed. We have revised the text in that section to clarify. 

Exceptional:

1. The open science implementation plan is amazing! I like that the authors are implementing what they believe in.

** Our Response: We are very gratified to see acknowledgement of our open science efforts. Thank you for this!

2. It is great that the authors deduplicate articles but still save all articles. Maybe both the original article and the most recent one could be saved?!

** Our Response: We don’t quite understand this comment. Do you mean drop all those between the first and the most recent? We saved all for the sake of replication and being able aggregate metrics (e.g., all citations, etc.). 

3. Thank you for disclosing the data issues faced.

** Our Response: Thank you for this as well. We were uncertain what to do at the time, and it’s validating to see transparency valued. 

Reviewer #3: 

The authors of the submitted manuscript report on a thorough work on producing an open catalogue of biodata resources. They describe the process of collecting and verifying information for such a catalogue. The authors used ML, NLP plus manual adjustments. The code and the catalogue itself are made available to the research community. While the used methods are straight forward, their usage is well justified and applying them to a large corpus of scientific publication is a time-consuming process. The manuscript is well written and structured, the reported results are of great importance. Therefore, I believe this manuscript should be accepted for a publication after minor revisions.

** Our Response: It was gratifying to see these very positive comments!

Suggestions for improvement of the current draft: [133 ‘used a machine-learning based approach’ It is not clear at the beginning of the manuscript what ml was employed? Is it using BERT? It should be explained from the beginning.

** Our Response: We have revised the text to introduce our use of BERT much earlier. Thank you for catching this omission. 

In my view, the manuscript could be shortened. For example, there is a bit of repetition in sections 2 and 3. Also, some parts could be moved to SI (supplementary information).

** Our Response: We did revise throughout to tighten the text and remove as much duplication as possible. It is not shorter due to suggestions from other reviewers; however, their suggestions were valuable and we think the additions strengthen the paper even whilst lengthening it. 

[555] ‘111 resources were funded by international agencies...’ It is a lot of resources. It is not clear why funding agencies that could not be mapped to a single country were not shown.

** Our Response: We extracted funding information for 1714 resources so that’s 111/1714 (6.5%), which is not trivial but relatively small. We’re not exactly sure how we would have mapped the multinational funders. For example, if we had tried to map all the individual countries associated with the EU, would we have included or excluded the UK as funding was both before and after Brexit? Or did the reviewer mean something else?

[623] ‘the majority of life science resources within both re3data (975/1189, 80.5%) and FAIRsharing 635 (1161/1640, 70.8%) are not found in our inventory’. That is a real worry. This warrants a more detailed investigation, explanation and suggestion for a resolution.

** Our Response: We absolutely agree that a more detailed investigation is warranted. We know from the preliminary work presented here, however, that this will certainly require a substantial amount of human curation and would constitute another study entirely. Our preliminary analysis suggests that none are entire subsets of each other, but without more careful investigation we don’t know the true extent of the overlap (or lack thereof), so we have edited the text to be less premature in our conclusions. Regardless, our initial results are surprising enough to justify the follow-up work required, and we’re glad there’s interest in seeing it!

Personally, I do not like the title: ‘A machine learning-enabled open biodata resource inventory from the scientific literature’. I suggest or to be more specific what ml was used, or drop mentioning of ml. I do not see why it is so important that ml was used to make it to the title. The important aspects are: open, re-usable, updatable, scientific literature driven.

** Our Response: We were very gratified to see the reviewers' strong appreciation for open science! Thank you! In regards to dropping ML from the title, we’d like to respectfully decline as ML was the main methodical approach so it seems important to forefront. We also think it would be good to keep the ML as the broad term to pique the interest of ML researchers and reinforce ML’s utility to life sciences readers. 

Just to add, it was interesting to see the results of Mid-Project evaluation and iteration.

** Our Response: Thank you!

6. PLOS authors have the option to publish the peer review history of their article (what does this mean?). 

Reviewer #1: No

Reviewer #2: No

Reviewer #3: Yes: Larisa Soldatova

** Our Response: We respect the reviewers’ choices, although we’d just like to say that it’s too bad that Reviewer 2 especially didn’t opt to be identified because they provided a very high quality review.

---

## [Decision Letter · Decision Letter 1]

29 Sep 2023

PONE-D-23-16263R1A machine learning-enabled open biodata resource inventory from the scientific literaturePLOS ONE

Dear Dr. Imker,

Thank you for submitting your manuscript to PLOS ONE. After careful consideration, we feel that it has merit but does not fully meet PLOS ONE’s publication criteria as it currently stands. Therefore, we invite you to submit a revised version of the manuscript that addresses the points raised during the review process.

We look forward to receiving your revised manuscript.

Kind regards,

Anna Bernasconi, PhD

Academic Editor

PLOS ONE

Journal Requirements:

Additional Editor Comments:

Dear authors, all reviewers and myself agree that the manuscript has much improved with your revision. Reviewer 2 makes some minor comments that can help finalize your paper, making it ready for acceptance. Please address them in a minor revision.

Reviewers' comments:

Reviewer's Responses to Questions

**Comments to the Author**

1. If the authors have adequately addressed your comments raised in a previous round of review and you feel that this manuscript is now acceptable for publication, you may indicate that here to bypass the “Comments to the Author” section, enter your conflict of interest statement in the “Confidential to Editor” section, and submit your "Accept" recommendation.

Reviewer #1: All comments have been addressed

Reviewer #2: (No Response)

Reviewer #3: All comments have been addressed

2. Is the manuscript technically sound, and do the data support the conclusions?

Reviewer #1: Yes

Reviewer #2: Yes

Reviewer #3: Yes

3. Has the statistical analysis been performed appropriately and rigorously? 

Reviewer #1: Yes

Reviewer #2: N/A

Reviewer #3: Yes

4. Have the authors made all data underlying the findings in their manuscript fully available?

Reviewer #1: Yes

Reviewer #2: Yes

Reviewer #3: Yes

5. Is the manuscript presented in an intelligible fashion and written in standard English?

Reviewer #1: Yes

Reviewer #2: Yes

Reviewer #3: Yes

6. Review Comments to the Author

Reviewer #1: Thank you for adressing the different suggestions and comments. The manuscript has greatly improved and is very much worth publishing.

Reviewer #2: I want to thank the reviewers for including my comments in their work. I do agree that the manuscript is more readable overall. In general, it seems the biggest difficulty with this paper is delineating the methods and the results section in part because of the mid-project evaluation. I do think the current delineation is much better and I have a few minor comments and places that can be cut out due to repetition to offer.

Minor comments (in order of appearance):

1. I am surprised by how short the training set curation took. That is great. It seems that the task was pretty simple for them to do which you may want to comment on in relation to the portion that gets manually reviewed based on the threshold for prediction. The idea being that manual review is quick and easy even if it needs to happen for almost half of them.

2. Section 3.3.1 (page 28): the paragraph after figure 6 is too long and mostly repeating the methods about the mid-project evaluation. The results are that the method chosen was the selectively mediated approach, the threshold of 0.978, and the sentences starting with "In the 468 articles from the 10% sample..." Potentially this paragraph should start with the last sentence because that sums up the main results for this section.

3. Section 3.3.1 (page 29): similarly to comment 2, the next paragraph starting with "During preparation of this manuscript, we realized..." is methods. The paragraph should start with "In the mid-project evaluation, 7 articles from the classification test and 5 articles from the NER test set were found..."

4. Section 3.3.2: There are a lot of records flagged for manual review (almost half if you add up all the different reasons for being flagged. Harkening back to comment 1, I think you can say that manual review is quick and easy so it doesn't matter that it is so much. It is also interesting to note the number that were flagged here vs. the test data if that is easy to do.

5. In general section 3.5 is a great example of just results and pretty short. I would still say that you can cut out part of the first sentence of section 3.5 starting with "After curation and processing to remove..." Instead the section can start with "The resulting inventory contained 3112 resources..."

6. Section 3.5.2: The first paragraph talks about the metadata and everything after "Both PubMed Central and Europe PMC, which exchange data,..." is introduction or discussion/future work. I don't think it belongs in results. Further the last paragraph in that section is also discussion/future work starting with "The articles associated with biodata resources within the inventory also provide opportunities to glean...". This whole paragraph is mostly stated as future work. If you want it to be results about the articles instead, then maybe providing an example of why they are useful or could be used would be helpful.

7. In the discussion section, the second paragraph states, "the selected article classification model achieved..." Please state the name of the models that performed the best for classification and NER.

8. In the discussion section, the added discussion on why the overlap between resources is low is great. One comment mentioned is that the selection criteria may differ in subtle ways. I guess can you give examples or do a very basic analysis of the differences?

9. In the discussion section (page 39). The sentence starting with "One potential use of the inventory as a catalyst for outreach." needs to start with "This is one potential use of the inventory..."

Response to authors comments on my review:

1. I see that you are using the validation set to choose the algorithm and then using the test set to report on unseen data. Thank you for the explanation.

2.Thank you for your comment about the "true negatives". It is very interesting to think about in theory vs. practice.

3. I hear the point about the manuscript being very long. I think mentioning more things in future work is totally fine. I do believe there is good reception of this work and that it is interesting.

4. I understand the authors' decision to keep the structure following PLOS One guidelines. Overall the structure is much clearer than before.

5. Minor revision comment 2: I was asking about the latter option of whether knowledge-bases and ontologies were included in your definition and find that interesting in general and maybe it should be mentioned as well. I like the discussion in the paper about how to define a biodata resource.

6. I do understand the difficulty with methods and results so take my comments around that with a grain of salt.

7. Adding F1 score makes sense when you do have two different metrics you use. Thank you for the response.

8. Thank you for the figure comments. All of that makes sense.

9. In terms of the deduplication comment I made in the exceptional category, I see that you do keep the articles so you are all good.

Reviewer #3: I am satisfied with the revisions made. The authors have addressed the comments and suggestions by other reviewers reasonably well.

7. PLOS authors have the option to publish the peer review history of their article (what does this mean?). If published, this will include your full peer review and any attached files.

Reviewer #1: No

Reviewer #2: **Yes: **Mayla R Boguslav

Reviewer #3: No

---

## [Author Response · Author response to Decision Letter 1]

3 Nov 2023

As provided in the "Response to Reviewers" documentment:

Reviewer Prompts/Comments

1. If the authors have adequately addressed your comments raised in a previous round

of review and you feel that this manuscript is now acceptable for publication, you may

indicate that here to bypass the “Comments to the Author” section, enter your conflict of

interest statement in the “Confidential to Editor” section, and submit your "Accept"

recommendation.

Reviewer #1: All comments have been addressed

Reviewer #2: (No Response)

Reviewer #3: All comments have been addressed

Our Response: No action needed.

2. Is the manuscript technically sound, and do the data support the conclusions?

Reviewer #1: Yes

Reviewer #2: Yes

Reviewer #3: Yes

Our Response: No action needed.

3. Has the statistical analysis been performed appropriately and rigorously?

Reviewer #1: Yes

Reviewer #2: N/A

Reviewer #3: Yes

Our Response: No action needed.

4. Have the authors made all data underlying the findings in their manuscript fully

available?

Reviewer #1: Yes

Reviewer #2: Yes

Reviewer #3: Yes

Our Response: No action needed.

5. Is the manuscript presented in an intelligible fashion and written in standard English?

Reviewer #1: Yes

Reviewer #2: Yes

Reviewer #3: Yes

Our Response: No action needed.

6. Review Comments to the Author

Reviewer #1

Thank you for addressing the different suggestions and comments. The

manuscript has greatly improved and is very much worth publishing.

Our Response: We are happy to hear so!

Reviewer #2:

I want to thank the reviewers for including my comments in their work. I do agree

that the manuscript is more readable overall. In general, it seems the biggest

difficulty with this paper is delineating the methods and the results section in part

because of the mid-project evaluation. I do think the current delineation is much

better and I have a few minor comments and places that can be cut out due to

repetition to offer.

Our Response: This is great for us to hear. Specific points are addressed below.

Minor comments (in order of appearance):

1. I am surprised by how short the training set curation took. That is great. It

seems that the task was pretty simple for them to do which you may want to

comment on in relation to the portion that gets manually reviewed based on the

threshold for prediction. The idea being that manual review is quick and easy

even if it needs to happen for almost half of them.

Our Response: Yes, relatively speaking, we found curation to be fairly

straightforward, which did make the mediated approach feasible. We revised the

text to make this point while presenting our rationale for doing the selective

mediated review.

2. Section 3.3.1 (page 28): the paragraph after figure 6 is too long and mostly

repeating the methods about the mid-project evaluation. The results are that the

method chosen was the selectively mediated approach, the threshold of 0.978,

and the sentences starting with "In the 468 articles from the 10% sample..."

Potentially this paragraph should start with the last sentence because that sums

up the main results for this section.

Our Response: We tried to revise as suggested, but the text was cumbersome

when placed elsewhere, in particular because it does contain results (e.g.,

precision vs F1 lowered recall to an unacceptable level for NER). This caused

the text to be quite disjointed, so we respectfully decline to implement this

suggestion.

3. Section 3.3.1 (page 29): similarly to comment 2, the next paragraph starting

with "During preparation of this manuscript, we realized..." is methods. The

paragraph should start with "In the mid-project evaluation, 7 articles from the

classification test and 5 articles from the NER test set were found..."

Our Response: We have revised this paragraph and moved some nonredundant

content to methods.

4. Section 3.3.2: There are a lot of records flagged for manual review (almost half

if you add up all the different reasons for being flagged. Harkening back to

comment 1, I think you can say that manual review is quick and easy so it doesn't

matter that it is so much. It is also interesting to note the number that were

flagged here vs. the test data if that is easy to do.

Our Response: We added this point explicitly in the method sections already per

suggestions above, so we respectfully decline as other suggested revisions

asked us to remove such explanatory text from the results sections.

5. In general section 3.5 is a great example of just results and pretty short. I

would still say that you can cut out part of the first sentence of section 3.5 starting

with "After curation and processing to remove..." Instead the section can start

with "The resulting inventory contained 3112 resources..."

Our Response: We respectfully decline this suggestion as we believe the

opening clause helps keep the reader oriented.

6. Section 3.5.2: The first paragraph talks about the metadata and everything

after "Both PubMed Central and Europe PMC, which exchange data,..." is

introduction or discussion/future work. I don't think it belongs in results. Further

the last paragraph in that section is also discussion/future work starting with "The

articles associated with biodata resources within the inventory also provide

opportunities to glean...". This whole paragraph is mostly stated as future work. If

you want it to be results about the articles instead, then maybe providing an

example of why they are useful or could be used would be helpful.

Our Response: Only the last half of the last sentence in this paragraph has

anything to do with future work, and even that was in direct relation to the results.

Our point about the irregular reporting of smaller funders (especially those

outside of the US and UK) directly precedes our results where we found

increasingly more granular funding sources reported. There is no good place in

the introduction or the discussion to bring this up and even if so, it would be too

far away from where this context is most helpful for the reader. We have edited

slightly, however, to make it more clear that we see this directly in the results of

our work.

7. In the discussion section, the second paragraph states, "the selected article

classification model achieved..." Please state the name of the models that

performed the best for classification and NER.

Our Response: We have edited to explicitly state the model names in the

discussion as well.

8. In the discussion section, the added discussion on why the overlap between

resources is low is great. One comment mentioned is that the selection criteria

may differ in subtle ways. I guess can you give examples or do a very basic

analysis of the differences?

Our Response: We have edited, and provide an example we point out that that

definition of “data” itself is not universally shared.

9. In the discussion section (page 39). The sentence starting with "One potential

use of the inventory as a catalyst for outreach." needs to start with "This is one

potential use of the inventory..."

Our Response: We have corrected the typo. Thank you for catching it.

Response to authors comments on my review:

1. I see that you are using the validation set to choose the algorithm and then

using the test set to report on unseen data. Thank you for the explanation.

Our Response: You’re welcome.

2.Thank you for your comment about the "true negatives". It is very interesting to

think about in theory vs. practice.

Our Response: You’re welcome.

3. I hear the point about the manuscript being very long. I think mentioning more

things in future work is totally fine. I do believe there is good reception of this

work and that it is interesting.

Our Response: Great to hear!

4. I understand the authors' decision to keep the structure following PLOS One

guidelines. Overall the structure is much clearer than before.

Our Response: Great to hear!

5. Minor revision comment 2: I was asking about the latter option of whether

knowledge-bases and ontologies were included in your definition and find that

interesting in general and maybe it should be mentioned as well. I like the

discussion in the paper about how to define a biodata resource.

Our Response: We did not restrict our definition to only experimental data, so

knowledge bases and ontologies could also meet the criteria and, in fact, did.

Two examples of ontologies in the inventory are FOAM (Functional Ontology

Assignments for Metagenomes) and ENVO (Environmental Ontology), while two

examples of KBs in the inventory are FROG-kb (Forensic Resource/Reference

on Genetics-knowledge base) and GPKB (Genomic and Proteomic Knowledge

Base). There may be others as well but we provide these just as examples to

know that ontologies and knowledge bases are included.

6. I do understand the difficulty with methods and results so take my comments

around that with a grain of salt.

Our Response: Excellent. Thank you!

7. Adding F1 score makes sense when you do have two different metrics you

use. Thank you for the response.

Our Response: You’re welcome.

8. Thank you for the figure comments. All of that makes sense.

Our Response: Great!

9. In terms of the deduplication comment I made in the exceptional category, I

see that you do keep the articles so you are all good.

Our Response: Great! Thank you again for your time and terrific attention to

detail.

Reviewer #3

I am satisfied with the revisions made. The authors have addressed the

comments and suggestions by other reviewers reasonably well.

Our Response: We are very glad to hear so!

7. PLOS authors have the option to publish the peer review history of their article (what does

this mean?). If published, this will include your full peer review and any attached files.

Our Response: Great to have 2 of 3 allow their names open! We appreciate all of their time

and very helpful comments.

---

## [Editor Report · Decision Letter 2]

8 Nov 2023

A machine learning-enabled open biodata resource inventory from the scientific literature

PONE-D-23-16263R2

Dear Dr. Imker,

We’re pleased to inform you that your manuscript has been judged scientifically suitable for publication and will be formally accepted for publication once it meets all outstanding technical requirements.

Kind regards,

Anna Bernasconi, PhD

Academic Editor

PLOS ONE

Additional Editor Comments (optional):

Dear authors, the paper has improved in the two revisions; all reviewers and myself agree that it can now be accepted for publication.
---

## [Editor Report · Acceptance letter]

16 Nov 2023

PONE-D-23-16263R2 

A machine learning-enabled open biodata resource inventory from the scientific literature 

Dear Dr. Imker:

I'm pleased to inform you that your manuscript has been deemed suitable for publication in PLOS ONE. Congratulations! Your manuscript is now with our production department. 

Kind regards, 

on behalf of

Dr. Anna Bernasconi 

Academic Editor

PLOS ONE